# Hidden GPCR structural transitions addressed by multiple walker supervised molecular dynamics (mwSuMD)

**Giuseppe Deganutti[1]\*, Ludovico Pipito[1], Roxana Maria Rujan[1], Tal Weizmann[1], Peter Griffin[1], Antonella Ciancetta[2], Stefano Moro[3], Christopher Arthur Reynolds[1,4]\***

[1]Centre for Health and Life Sciences, Coventry University, Coventry, United Kingdom; [2]Dipartimento di Scienze Chimiche, Farmaceutiche ed Agrarie, University of Ferrara, Ferrara, Italy; [3]Molecular Modeling Section (MMS), Dipartimento di Scienze del Farmaco, University of Padua via Marzolo, Padova, Italy; [4]School of Life Sciences, University of Essex, Wivenhoe Park, Colchester, United Kingdom

## eLife Assessment

This study describes an improved adaptive sampling approach, multiple-walker Supervised Molecular Dynamics (mwSuMD), and its application to G protein-coupled receptors (GPCRs), which are the most abundant membrane proteins and key targets for drug discovery. The manuscript provides **solid** evidence that the mwSuMD approach can assist in the sampling of complex binding processes, leading to **useful** findings for GPCR activity, including resolution of interactions not seen experimentally. The method has the potential to have broad applicability in structural biology and pharmacology.

**\*For correspondence:**
ad5288@coventry.ac.uk (GD);
ad5291@coventry.ac.uk (CAR)

**Competing interest:** The authors declare that no competing interests exist.

## Abstract

The structural basis for the pharmacology of human G protein-coupled receptors (GPCRs), the most abundant membrane proteins and the target of about 35% of approved drugs, is still a matter of intense study. What makes GPCRs challenging to study is the inherent flexibility and the metastable nature of interaction with extra- and intracellular partners that drive their effects. Here, we present a molecular dynamics (MD) adaptive sampling algorithm, namely multiple walker supervised molecular dynamics (mwSuMD), to address complex structural transitions involving GPCRs without energy input. We first report the binding and unbinding of the vasopressin peptide from its receptor $V_2$. Successively, we present the complete transition of the glucagon-like peptide-1 receptor (GLP-1R) from inactive to active, agonist and $G_s$-bound state, and the guanosine diphosphate (GDP) release from $G_s$. To our knowledge, this is the first time the whole sequence of events leading from an inactive GPCR to the GDP release is simulated without any energy bias. We demonstrate that mwSuMD can address complex binding processes intrinsically linked to protein dynamics out of reach of classic MD.

## Introduction

Supervised molecular dynamics (*Cuzzolin et al., 2016*; *Deganutti et al., 2020b*) (SuMD) is an efficient adaptive sampling technique for studying ligand-receptor binding and unbinding pathways; here, we present the multiple walker enhancement (mwSuMD) to study a significantly wider range of structural transitions relevant to the drug design. We validated the method by applying it to G protein-coupled receptors (GPCRs), since their inherent flexibility is essential to their function and because these are

the most abundant family of membrane receptors in eukaryotes (*de Mendoza et al., 2014*) and the target for more than one-third of drugs approved for human use (*Sriram and Insel, 2018*).

Vertebrate GPCRs are subdivided into five subfamilies (Rhodopsin or class A, Secretin or class B, Glutamate or class C, Adhesion, and Frizzled/Taste2) according to function and sequence (*Schiöth and Fredriksson, 2005*; *G protein-coupled receptors, 2019*). X-ray and cryo-electron microscopy (cryo-EM) show that GPCRs possess seven transmembrane (TM) helices connected by three extracellular loops (ECLs) and three intracellular loops (ICLs), with an extended and structured N-terminal extracellular domain (ECD) in all subtypes, but class A. The primary function of GPCRs is transducing extracellular chemical signals into the cytosol by binding and activating four G protein families ($G_{s/olf}$, $G_{i/o}$, $G_{12/13}$, and $G_{q/11}$) responsible for decreasing ($G_{i/o}$) or increasing ($G_{s/olf}$) cyclic adenosine-3′,5′-monophosphate, and generating inositol-1,4,5-triphosphate and diacylglycerol to increase $Ca^{2+}$ intracellular levels ($G_q$) (*Syrovatkina et al., 2016*).

Many aspects of GPCR pharmacology remain elusive: e.g., the structural determinants of the selectivity displayed toward specific G proteins or the ability of certain agonists to drive a preferred intracellular signaling pathway over others (i.e. functional selectivity or bias) (*Tan et al., 2018*). GPCRs are challenging proteins to characterize experimentally due to their inherent flexibility and the transitory nature of the complexes formed with extracellular and intracellular effectors. Importantly, agonists can allosterically modify the receptor selectivity profile by imprinting unique intracellular conformations from the orthosteric binding site. The mechanism behind these phenomena is one of the outstanding questions in the GPCR field (*Flock et al., 2017*).

Molecular dynamics (MD) is a powerful computational methodology that predicts the movement and interactions of (bio)molecules in systems of variable complexity, at atomic detail. However, classic MD sampling is limited to the microsecond or, in the best conditions, the millisecond time scale (*Hollingsworth and Dror, 2018*; *Durrant and McCammon, 2011*). For this reason, different algorithms have been designed to speed up the simulation of rare events such as ligand (un)binding and conformational transitions. Among the most popular and effective ones are metadynamics (*Bussi and Laio, 2020*) and path collective variables metadynamics (*D'Amore et al., 2024*), accelerated MD (aMD) (*Hamelberg et al., 2004*), and Gaussian-accelerated MD (GaMD) (*Miao et al., 2015*), which introduce an energy potential to overcome the energy barriers preventing the complete exploration of the free energy surface, thus de facto bias the simulation. Energetically unbiased MD protocols, on the other hand, comprise weighted ensemble MD (*Zuckerman and Chong, 2017*), swarms approach (*Fleetwood et al., 2020*), AdaptiveGoal (*Lovera et al., 2019*), and SuMD (*Cuzzolin et al., 2016*; *Sabbadin and Moro, 2014*), which have largely been applied to (unbinding) small molecules, peptides, and small proteins (*Cuzzolin et al., 2016*; *Sabbadin and Moro, 2014*; *Deganutti et al., 2021a*; *Dong et al., 2020*; *Deganutti et al., 2021c*; *Cary et al., 2022*). Since SuMD is optimized for (un)bindings, we have designed mwSuMD to address more complex conformational transitions and protein-protein associations. GPCRs preferentially couple to very few G proteins out of 23 possible counterparts (*Flock et al., 2017*; *Wall et al., 2022*). It is increasingly accepted that dynamic and transient interactions determine whether the encounter between a GPCR and a G protein results in productive or unproductive coupling (*Culhane et al., 2022*). MD simulations are considered a useful orthogonal tool for providing working hypotheses and rationalizing existing data on G protein selectivity. However, so far, it has not delivered as expected. Attempts have usually employed energetically biased simulations, have been confined to the Gα subunit, or considered a preformed GPCR:G protein complex (*Wall et al., 2022*; *Mattedi et al., 2020*; *Miao and McCammon, 2018*; *Mafi et al., 2023*).

First, we validated mwSuMD on the nonapeptide arginine vasopressin (AVP) by simulating binding (dynamic docking) and unbinding paths from the vasopressin 2 receptor ($V_2R$). Dynamic docking, although more computationally demanding than standard molecular docking, provides insights into the binding mode of ligands in a fully hydrated and flexible environment. Moreover, it informs about binding paths and the complete mechanism of formation leading to an intermolecular complex, delivering in the context of binding kinetics *Guo et al., 2017* and structure-kinetics relationship studies (*Guo et al., 2015*). We then studied the class B1 GPCR glucagon-like peptide-1 receptor (GLP-1R) activation by the small molecule PF06882961. GLP-1R is a validated target in type 2 diabetes and probably the best-characterized class B1 GPCR from a structural perspective. GLP-1R is the only class B1 receptor with structurally characterized non-peptidic orthosteric agonists, which makes it a model system for studying the druggability of the entire B1 subfamily. After GLP-1R agonist binding and

activation, the coupling of $G_s$ and the release of guanosine diphosphate (GDP), the rate-limiting step of the G protein activation, was simulated for the first time using an energy-unbiased method.

These results demonstrate the usefulness of mwSuMD for illuminating the molecular events involved in GPCR function.

## Results

### Short mwSuMD time windows improve the AVP dynamic docking prediction

AVP is an endogenous hormone (*Figure 1a*) that mediates antidiuretic effects on the kidney by signaling through three class A GPCR subtypes: $V_{1a}$ and $V_{1b}$ receptors activate phospholipases via $G_{q/11}$, while the $V_2R$ activates adenylyl cyclase by interacting with $G_s$ (*Birnbaumer, 2000*) and is a therapeutic target for hyponatremia, hypertension, and incontinence (*Ball, 2007*). AVP is amphipathic and in the bound state interacts with both polar and hydrophobic $V_2R$ residues located on TM helices and ECLs (*Figure 1b*). Although AVP presents an intramolecular C1-C6 disulfide bond that limits the backbone's overall conformational flexibility, it has many rotatable bonds, making dynamic docking complicated (*Gioia et al., 2017*). We compared the performance of mwSuMD to the parent algorithm SuMD in reconstructing the experimental $V_2R$:AVP complex using different settings, simulating a total of 92 binding events (*Table 1*). As a reference, the AVP root mean square deviation (RMSD) during a classic (unsupervised) equilibrium MD simulation of the X-ray AVP:$V_2R$ complex was 3.80±0.52 Å (*Figure 1—figure supplement 1*). SuMD (*Cuzzolin et al., 2016*; *Sabbadin and Moro, 2014*) produced a minimum RMSD to the cryo-EM complex of 4.28 Å, with most of the replicas (i.e. distribution's mode) having an RMSD close to 10 Å (*Figure 1—figure supplement 2a*). mwSuMD, with the same settings (*Figure 1—figure supplement 2b*, *Table 1*) in terms of time window duration (600 ps), metric supervised (the distance between AVP and $V_2R$), and acceptance method (slope) produced slightly more precise results (i.e. distribution's mode RMSD = 7.90 Å) but similar accuracy (minimum RMSD = 4.60 Å). Supervising the AVP RMSD to the experimental complex rather than the distance (*Figure 1—figure supplement 2c*) and using the SMscore (*Equation 1*) as the acceptance method (*Figure 1—figure supplement 2d*) worsened the prediction. Supervising distance and RMSD at the same time (*Figure 1—figure supplement 2e*), employing the DMscore (*Equation 2*), recovered accuracy (minimum RMSD = 4.60 Å) but not precision (distribution mode RMSD = 12.40 Å). Interestingly, decreasing the time window duration from 600 ps to 100 ps impaired the SuMD ability to predict the experimental complex (*Figure 1a*), but enhanced mwSuMD accuracy and precision (*Figure 1b–d*). The combination of RMSD as the supervised metric and SMscore produced the best results in terms of minimum RMSD and distribution mode RMSD, 3.85 Å and 4.40 Å, respectively (*Figure 1d*, *Video 1*), in agreement with the AVP deviations in the equilibrium MD simulation of the X-ray AVP:$V_2R$ complex (*Figure 1—figure supplement 1*). These results confirm the inherent complexity of reproducing the AVP:$V_2R$ complex via dynamic docking and suggest that short time windows can improve mwSuMD performance on this system. However, it is necessary to know the final bound state to employ the RMSD as the supervised metric, while the distance is required to dynamically dock ligands with unknown bound conformation as previously reported (*Cuzzolin et al., 2016*; *Wall et al., 2022*). Both distance and RMSD-based simulations delivered insights into the binding path and the residues involved along the recognition route. For example, mwSuMD pinpointed $V_2R$ residues E184[ECL2], P298[ECL3], and E303[ECL3] (*Figure 2a*) as involved during AVP binding, although not in contact with the ligand in the orthosteric complex. None of them are yet characterized through mutagenesis studies according to the GPCRdb (*Isberg et al., 2016*).

Further to binding, an SuMD approach was previously employed to reconstruct the unbinding path of ligands from several GPCRs (*Deganutti et al., 2020b*; *Atanasio et al., 2020*). We assessed mwSuMD's capability to simulate AVP unbinding from $V_2R$. Five mwSuMD and five SuMD replicas were collected using 100 ps time windows (*Table 1*). Overall, mwSuMD outperformed SuMD in terms of time required to complete a dissociation (*Figure 2c*, *Video 2*), producing dissociation paths almost 10 times faster than SuMD. Such rapidity in dissociating inherently produces a limited sampling of metastable states along the pathway, which can be compensated by seeding classic (unsupervised) MD simulations from configurations extracted from the unbinding pathway (*Deganutti et al., 2021b*; *Deganutti et al., 2020a*). Here, the $V_2R$ residues involved during the dissociation were comparable

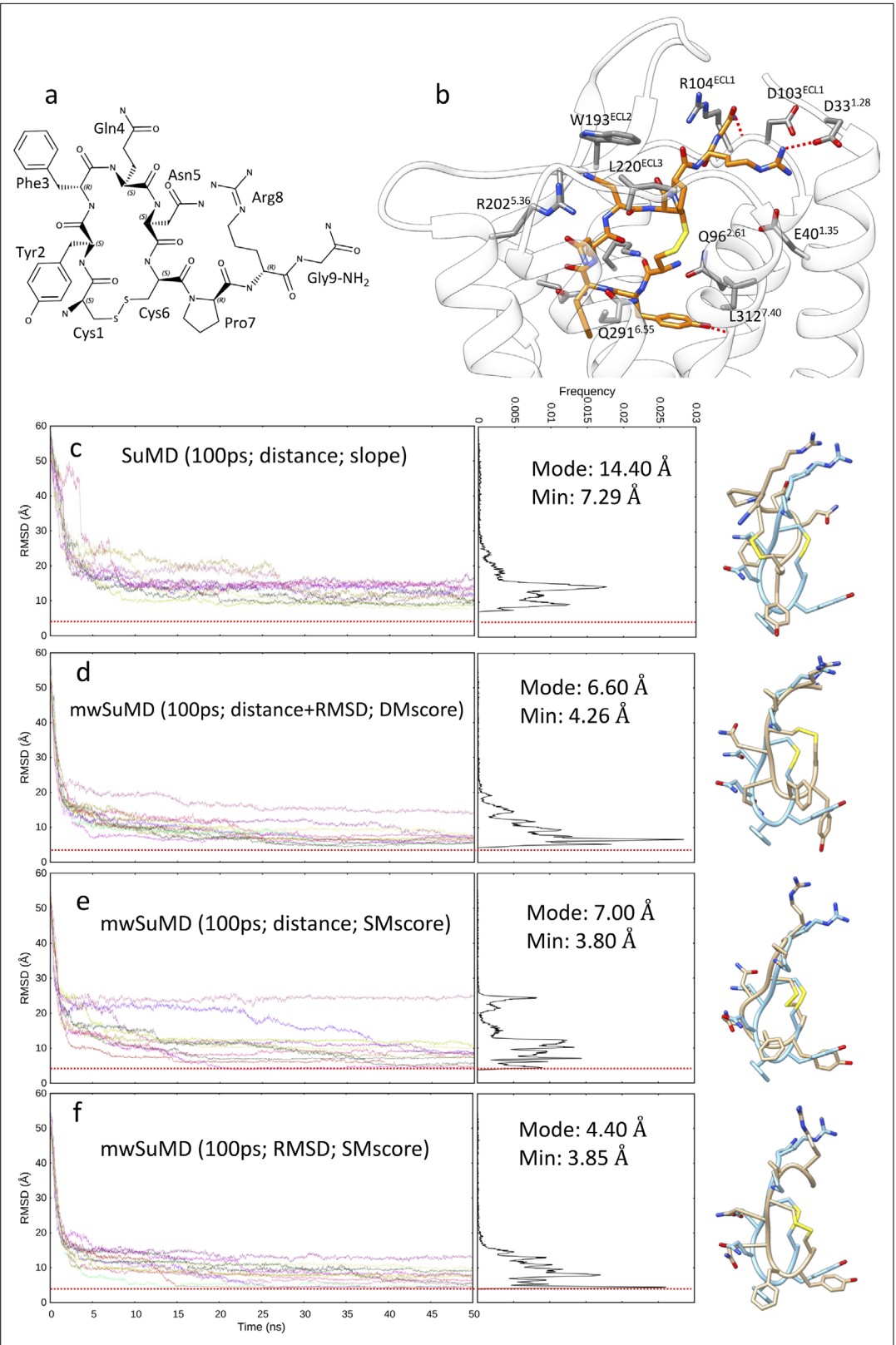

**Figure 1.** Arginine vasopressin (AVP) supervised molecular dynamics (SuMD) and multiple walker SuMD (mwSuMD) binding simulations to vasopressin 2 receptor (V₂R) (100 ps time windows). (**a**) Chemical structure and (**b**) binding more within the V₂R orthosteric binding site; AVP is represented in orange stick, while V₂R is in white ribbon and gray stick. For each set of settings (**c–f**) the root mean square deviation (RMSD) of AVP Cα atoms to the cryo-

*Figure 1 continued on next page*

*Figure 1 continued*

electron microscopy (cryo-EM) structure 7DW9 is reported during the time course of each SuMD (**c**) or mwSuMD (**d–f**) replica alongside the RMSD values distribution and the snapshot corresponding to the lowest RMSD values (AVP from the cryo-EM structure 7DW9 is in a cyan stick representation, while AVP from simulations is in a tan stick representation). A complete description of the simulation settings is reported in *Table 1* and the Methods section. The dashed red line indicates the AVP RMSD during a classic (unsupervised) equilibrium MD simulation of the X-ray AVP:V$_2$R complex (*Figure 1—figure supplement 1*).

The online version of this article includes the following figure supplement(s) for figure 1:

**Figure supplement 1.** Classic molecular dynamics (MD) simulation of the AVP:V$_2$R complex (PDB ID 7DW9).

**Figure supplement 2.** Arginine vasopressin (AVP) supervised molecular dynamics (SuMD) and multiple walker SuMD (mwSuMD) binding simulations to vasopressin 2 receptor (V$_2$R) (600 ps time windows).

---

to the binding (*Figure 2a and b*), although ECL2 and ECL3 were slightly more involved during the association than the dissociation, in analogy with other class A and B GPCRs (*Dong et al., 2020*; *Deganutti et al., 2021b*).

## PF06882961 binding and GLP-1R activation

The GLP-1R has been captured by cryo-EM in both the inactive and the active (G$_s$-bound) conformations and in complex with either peptide or non-peptide agonists (*Zhao et al., 2020*; *Kawai et al., 2020*; *Ma et al., 2020*; *Zhang et al., 2020*; *Cong et al., 2021*; *Cong et al., 2022b*). In the inactive GLP-1R, residues forming the binding site for the non-peptide agonist PF06882961 are dislocated

---

**Table 1.** Summary of all the simulations performed and the settings employed.

| System | # Replicas | TW duration | # Walkers | Metric | Acceptance |
|---|---|---|---|---|---|
| V$_2$R:AVP complex | 1 classic MD | 500 ns | N/A | N/A | N/A |
| | 9 (SuMD) | 600 ps | N/A | Distance | Slope |
| | 10 (mwSuMD) | 600 ps | 3 | Distance | Slope |
| | 11 (mwSuMD) | 600 ps | 3 | RMSD | Slope |
| | 8 (mwSuMD) | 600 ps | 3 | Distance | SMscore |
| | 9 (mwSuMD) | 600 ps | 3 | Distance and RMSD | DMscore |
| | 12 (mwSuMD) | 100 ps | 10 | Distance and RMSD | DMscore |
| | 11 (mwSuMD) | 100 ps | 10 | Distance | SMscore |
| | 11 (mwSuMD) | 100 ps | 10 | RMSD | SMscore |
| V$_2$R:AVP binding | 10 (SuMD) | 100 ps | N/A | Distance | Slope |
| V$_2$R:AVP unbinding | 5 (SuMD) | 100 ps | N/A | Distance | Slope |
| | 5 (mwSuMD) | 100 ps | 10 | Distance | SMscore |
| β$_2$ AR:G$_s$ protein binding | 3 (mwSuMD) | 100 ps | 5 | Distance and RMSD | DMscore |
| A1R:G$_i$ binding | 1 (mwSuMD) | 100 ps | 3 | RMSD | SMscore |
| GLP-1R:PF06882961 | 1 (mwSuMD) | 100 ps | 5 | Distance or RMSD (or a combination) | SMscore or DMscore |
| GLP-1R:G$_s$ protein binding | 3 (mwSuMD) | 200 ps | 3 | Distance or RMSD | SMscore or DMscore |
| G$_s$ AHD opening | 1 (mwSuMD) | 100 ps | 3 | Distance | SMscore |
| GLP-1R:G$_s$ GDP unbinding | 3 | 50 ps | 5 | Distance | SMscore |

N/A: not applicable; SuMD was not performed.

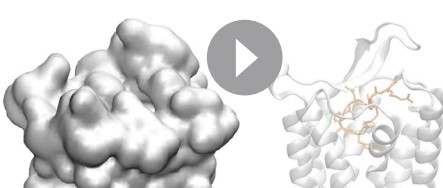

**Video 1.** Arginine vasopressin (AVP) binding (dynamic docking) to vasopressin 2 receptor (V₂R). Two-view of the mwSuMD replica better reproducing the AVP:V2R experimental complex. Left side: V2R is represented in white quick surface, while AVP is in transparent quick surface and green stick; right side: V2R is represented in white ribbon and cyan stick, while AVP is a green stick. The bound AVP conformation from PDB 7DW9 is reported as a reference in transparent orange ribbon and stick.

https://elifesciences.org/articles/96513/figures#video1

and scattered due to the structural reorganization of the transmembrane domain (TMD) and ECD (*Figure 3—figure supplement 1a*) that occurs on activation. Moreover, GLP-1R in complex with GLP-1 or different agonists present distinct structural features, even among structurally related ligands (*Figure 3—figure supplement 1b and c*). This complicates the scenario and suggests divergent recognition mechanisms among different agonists. We simulated the binding of PF06882961, reaching an RMSD to its bound conformation in 7LCJ of 3.79±0.83 Å (computed on the second half of the merged trajectory, superimposing on GLP-1R Cα atoms of TMD residues 150–390), using multistep supervision on different system metrics (*Figure 3*) to model the structural hallmark of GLP-1R activation (*Videos 3 and 4*).

Several metrics were supervised consecutively. First, the distance between PF06882961 and the TMD as well as the RMSD of the ECD to the active state (stage 1); second, the RMSD of ECD and ECL1 to the active state (stage 2); third, the RMSD of PF06882961 and ECL3 to the active state (stage 3); lastly, only the RMSD of TM6 (residues I345-F367, Cα atoms) to the active state (stage 4). The combination of these supervisions produced a conformational transition of GLP-1R toward the active state (*Figure 3*, *Video 4*). Noteworthy, mwSuMD, like any other CV-based technique, requires some knowledge of the simulated system. The sequence of these supervisions was arbitrary and does not necessarily reflect the right order of the steps involved in GLP-1R activation. This kind of planned multistep approach is feasible when the end-point receptor inactive and active structures are available, and the inherent flexibility of different domains is known. In class B GPCRs, the ECD is the most dynamic substructure, followed by the ECL1 and ECL3, which display high plasticity during ligand binding (*Dong et al., 2020*; *Cong et al., 2022a*). For this reason, we first supervised these elements of GLP-1R, leaving the bottleneck of activation, TM6 outward movement, as the last step. However, the protocol employed can be tweaked to study how each conformational transition takes place and influences the receptor domains. Structural elements not directly supervised, such as TM1 or TM7, were influenced by the movement of supervised helices or loops and therefore displayed an RMSD reduction to the active state. For example, the supervision of ECL3 (stage 3) and TM6 (stage 4) facilitated the spontaneous rearrangement of the ECD to an active-like conformation after the ECD had previously experienced transient high flexibility during stages 2 and 3 (*Figure 3*). These results suggest a concerted conformational transition for ECD and ECL1 during the binding of PF06882961 and an allosteric effect between ECL3 and the bottom of TM6. While the intracellular polar interactions were destabilized by the ECL3 transition to an active-like conformation (stages 2 and 3), the outward movement of TM6 (stage 4) did not favor the closure of ECL3 toward PF06882961, which appears to be driven by direct interactions between the ligand and R310[5.40] or R380[7.35]. Interestingly, the mwSuMD simulation during stage 4 (TM6 supervision) sampled a counterclockwise helix rotation (*Figure 3—figure supplement 2a*) consistent with the GLP-1R cryo-EM structures in the active, Gₛ-coupled state (*Zhang et al., 2020*; *Zhang et al., 2021*).

It is worth noting that 6LN2, the only inactive GLP-1R structure available complete with the ECD, was stabilized and solved with an antibody bound to the ECD. This strategy might have forced the ECD into a closed conformation that engages the EC *vestibule* of GLP-1R and possibly restrained the whole TMD in an altered conformation that deviates from the physiological conditions. This might explain why the RMSD of the TMD elements monitored during the simulation rarely reach values lower than 3 Å or 4 Å.

During the supervision of ECL3 and PF06882961 (stage 3), we observed a loosening of the intracellular polar interactions that stabilize GLP-1R TM6 in the inactive state. As a result, the subsequent

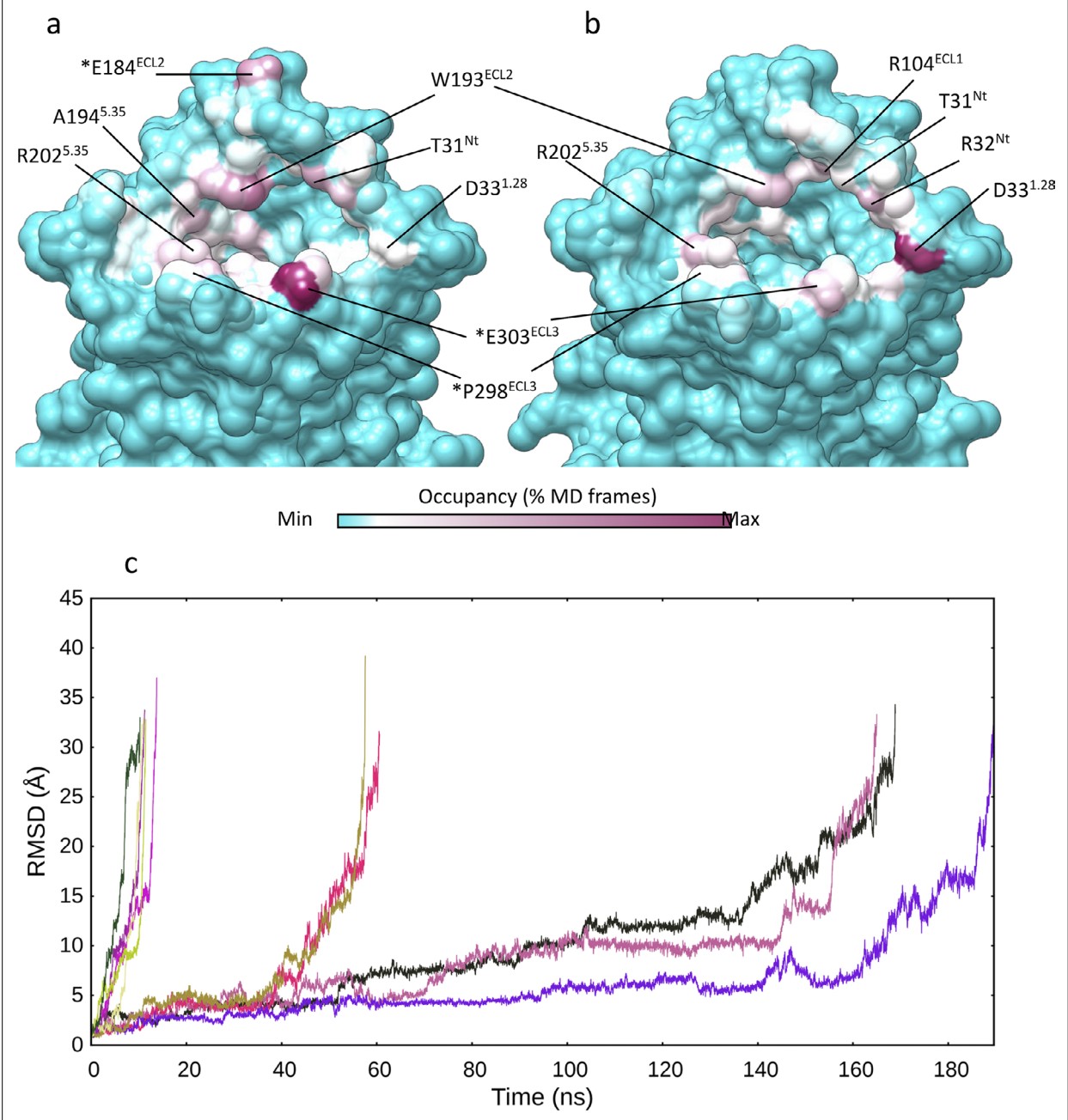

**Figure 2.** Multiple walker supervised molecular dynamics (mwSuMD) (un)binding simulations of arginine vasopressin (AVP). Vasopressin 2 receptor (V₂R) residues involved in mwSuMD simulations of AVP. (**a**) Binding (dynamic docking); (**b**) unbinding simulations. (**c**) Dissociation of AVP from V₂R. The root mean square deviation (RMSD) of AVP to the initial bound state is reported during the time course of five replicas of SuMD and mwSuMD, respectively.

supervision of TM6 (residues I345-F367, Cα atoms) rapidly produced the outward movement of TM6 toward the active state, in the last step of the mwSuMD simulation (stage 4). A more detailed analysis revealed that the central polar network, which is pivotal for mediating GLP-1 signaling (*Wootten et al., 2016*), and the residues at the TM6 kink level adopted active-like conformations during the final stage of the simulation (*Figure 4a*). In particular, the central polar network (E364^6.53, H363^6.52, and Q394^7.49) experienced side chain rearrangements (*Figure 4b*) while extensive TM6 kink dislocation occurred at L360^6.49, P358^6.48, L357^6.47, and I357^6.46 (*Figure 4c*).

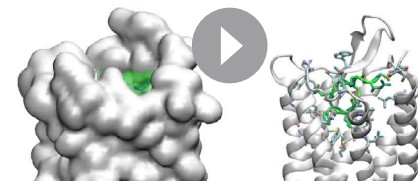

**Video 2.** Arginine vasopressin (AVP) unbinding from vasopressin 2 receptor (V₂R). Two-view of the five mwSuMD replicas performed. Left side: V2R is represented in white quick surface, while AVP is in transparent quick surface and green stick; right side: V2R is represented in white ribbon and cyan stick, while AVP is a green stick.

https://elifesciences.org/articles/96513/figures#video2

## Gₛ protein binding to GLP-1R and GDP release

We then focused on simulating the Gₛ binding to GLP-1R, after activation, *without energy input*. For this purpose, we first tested the binding between the prototypical class A receptor β₂ adrenoreceptor (β₂ AR) and the stimulatory G protein (Gₛ) (*Video 5*, *Figure 5—figure supplement 1a and b*) by supervising the distance between Gₛ helix 5 (α5) and β₂ AR as well as the RMSD of the intracellular end of TM6 to the fully active state of the receptor (see Supplementary Methods). During two out of three replicas, both Gα and Gβ achieved distance values close to 5 Å (minimum RMSD = 3.94 Å and 3.96 Å respectively), in good agreement with the reference (the β₂ AR:Gₛ complex, PDB 3SN6, *Figure 5—figure supplement 1c*). A possible pitfall is that G proteins bear potential palmitoylation and myristoylation sites that anchor the inactive trimer to the plasma membrane (*Linder et al., 1993*; *Zhang et al., 2004*), de facto restraining possible binding paths to the receptor. To address this point and test different conditions, we prepared the adenosine A1 receptor (A₁R) and its principal effector, the inhibitory G protein (Gᵢ) considering the Gᵢα residue C3 and Gγ residue C65 as palmitoylated and geranylgeranylated respectively and hence inserted in the membrane. Recently, the Gᵢ binding to A₁R was simulated by combining the biased methods GaMD with SuMD (*Li et al., 2022*) but without considering membrane-anchoring posttranslational modifications. Both classic (unsupervised) and mwSuMD simulations were performed on this system for comparison (*Video 6*, *Figure 5—figure supplement 1d*). In about 50 ns of mwSuMD, the Gᵢα subunit engaged its intracellular binding site on A₁R and formed a complex in good agreement with the cryo-EM structure (PDB 6D9H, RMSD ≈ 5 Å). For comparison, 1 µs of cMD did not produce a productive engagement as the Gᵢα remained at RMSD values >40 Å (*Figure 5—figure supplement 1d*), suggesting the effectiveness of mwSuMD in sampling G protein binding rare events without the input of energy. The membrane anchoring affected the overall Gᵢ binding and the final complex, which was rotated compared to the experimental structure due to the lipidation of Gᵢα and Gγ (*Figure 5—figure supplement 1e*).

Encouraged by results obtained on Gₛ and Gᵢ binding to β₂ AR and A₁R (*Figure 5—figure supplement 1*), we extracted the GLP-1R active conformation described above and simulated the Gₛ binding to its intracellular side. Starting from the inactive, membrane-anchored Gₛ, we performed three independent mwSuMD replicas by supervising the distance between Gₛ helix α5 and GLP-1R residues located at the intracellular binding interface (*Figure 5a and e*). All three mwSuMD replicas showed the Gₛ approaching GLP-1R, with two out of three reaching an RMSD of the Gₛα subunit close to or less than 10 Å, compared to the experimental complex 7LCI (*Figure 5e*). Replica 2, in particular, reproduced the cryo-EM GLP-1R:Gₛ complex with RMSD values to 7LCI of 7.59±1.58 Å, 12.15±2.13 Å, and 13.73±2.24 Å for Gα, Gβ, and Gγ, respectively. Such values do not support convergence with the static experimental structure but are not far from the RMSDs measured in our previous simulations of GLP-1R in complex with Gₛ and GLP-1 (*Deganutti et al., 2022*) (Gα=6.18 ± 2.40 Å; Gβ=7.22 ± 3.12 Å; Gγ=9.30 ± 3.65 Å), which indicates overall higher flexibility of Gβ and Gγ compared to Gα, which acts as a sort of fulcrum bound to GLP-1R.

According to the model of G protein activation, the G protein binding has the effect of allosterically stabilizing the orthosteric agonist in complex with the receptor (*Deganutti et al., 2022*) and destabilizing the GDP bound to Gα, triggering its release and exchange with guanosine triphosphate (*Gregorio et al., 2017*), upon opening of the G protein alpha-helical domain (AHD). Following this model, PF06882961 and GDP were respectively stabilized and destabilized during the simulated Gₛ association (*Figure 3—figure supplement 2b and c*). The analysis of atomic contacts along the binding path of Gₛ to GLP-1R highlights a few persistent interactions not observed in the equilibrium MD simulations of GLP-1R:Gₛ cryo-EM complexes *Deganutti et al., 2022*; e.g., we propose the

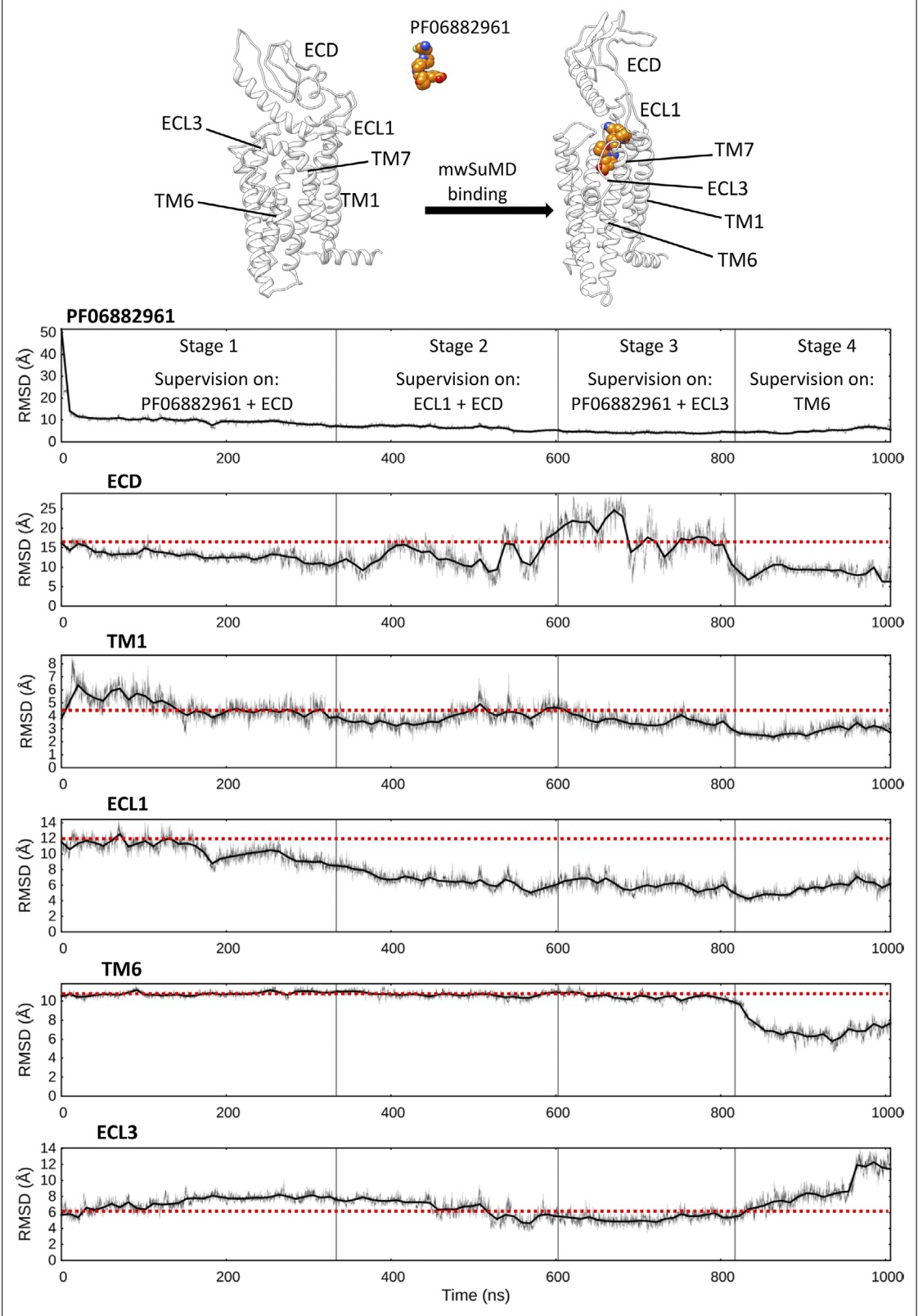

**Figure 3.** Multiple walker supervised molecular dynamics (mwSuMD) simulation of PF06882961 binding to glucagon-like peptide-1 receptor (GLP-1R) and receptor activation. Each panel reports the root mean square deviation (RMSD) to the position of the ligand in the active state (top panel) or a GLP-1R structural element over the time course (all but ECL3 converging to the active state). ECD: extracellular domain; TM: transmembrane helix; ECL:

*Figure 3 continued on next page*

*Figure 3 continued*

extracellular loop. The mwSuMD simulation was performed with four different settings over 1 μs in total. The red dashed lines show the initial RMSD value for reference.

The online version of this article includes the following figure supplement(s) for figure 3:

**Figure supplement 1.** Multiple walker supervised molecular dynamics (mwSuMD) simulation of PF06882961, glucagon-like peptide-1 receptor (GLP-1R) activation, and guanosine diphosphate (GDP) release.

**Figure supplement 2.** Dynamic feautures of GLP-1R activation and Gs binding.

hidden interaction between D344$^{6.33}$ and R385$^{\alpha5}$ to be important for G$_s$ coupling (*Appendix 1—table 1*), which would explain the GLP-1 EC$_{50}$ reduction upon mutation of position 344 to Ala (*Yuan et al., 2023*).

Extending Replica 2, we further investigated the G$_s$ activation mechanism by supervising the opening of the G$_s$ AHD, which is considered a necessary step to allow GDP release from the Ras-like domain (*Dror et al., 2015*). We first easily obtained the opening of AHD (*Figure 5b*) and successively supervised the GDP unbinding in a further three replicas, seeded after the AHD opening. In one of these three mwSuMD simulations, the nucleotide dissociated from G$_s$ (*Figure 5d*). *Video 7* shows the full G$_s$ binding**,** AHD opening, and GDP release. mwSuMD suggested several structural changes as implicated in GDP dissociation (*Figure 5f–h*): (i) the AHD opening; (ii) the conformational change of αG-α4 loop (residues A303$^{G\alpha}$-P332$^{G\alpha}$), in concert with a loosening of interactions between D323$^{G\alpha}$ and K342$^{6.31}$, and (iii) the rupture of the hydrogen bond between GDP and D295$^{G\alpha}$ triggered by the movement of αG away from the GDP binding site (*Figure 5f*). The involvement of αG through the αG-α4 during the release of GDP from G$_s$ is supported by hydrogen/deuterium exchange experiments (*Du et al., 2019*), while the role of D275$^{G\alpha}$ has been probed with functional assays on the G$_i$ isoform mutant D272$^{G\alpha i}$A (*Ham et al., 2021*). Moreover, a similar αG behavior was recently suggested by MD simulations of the G$_s$ binding pathway to β$_2$ AR (*Batebi et al., 2024*). We also note that the αG-α4 loop length and amino acidic composition diverge among G protein isoforms, further suggesting a role in G protein selectivity consistent with the hypothesis that, in different G proteins, distinct domains of the Gα subunit could be responsible for receptor selectivity (*Glukhova et al., 2018*). However, a more subtle allosteric communication through internal structural elements like the β2-β3 strands, prompted by α5 tilting, could have weakened GDP phosphate binding as previously suggested by other groups (*Sun et al., 2018*; *Kaya et al., 2016*; *Flock et al., 2015*). Interestingly, no significant conformational changes of the β6-α5 loop happened before or during the GDP dissociation, suggesting that its conformational change as captured in the nucleotide-free GLP-1R:G$_s$ complex (*Figure 5—figure supplement 2*) occurs after the GDP release as a result of the loss of binding stabilization, rather than being an initiator of the GDP dissociation.

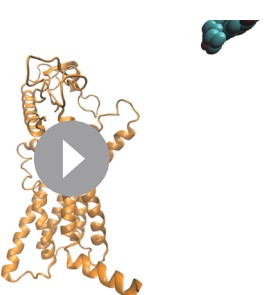

**Video 3.** Morph of the PF06882961 binding and glucagon-like peptide-1 receptor (GLP-1R) activation. The first and last frames of the PF06882961 binding simulations have been interpolated to produce a smoothed representative transition.

https://elifesciences.org/articles/96513/figures#video3

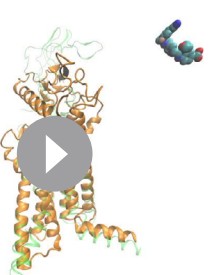

**Video 4.** PF06882961 binding and glucagon-like peptide-1 receptor (GLP-1R) activation. PF06882961 is represented in cyan van der Waals spheres, while GLP-1R is orange ribbon; the GLP-1R experimental active conformation is reported in transparent green ribbon as reference.

https://elifesciences.org/articles/96513/figures#video4

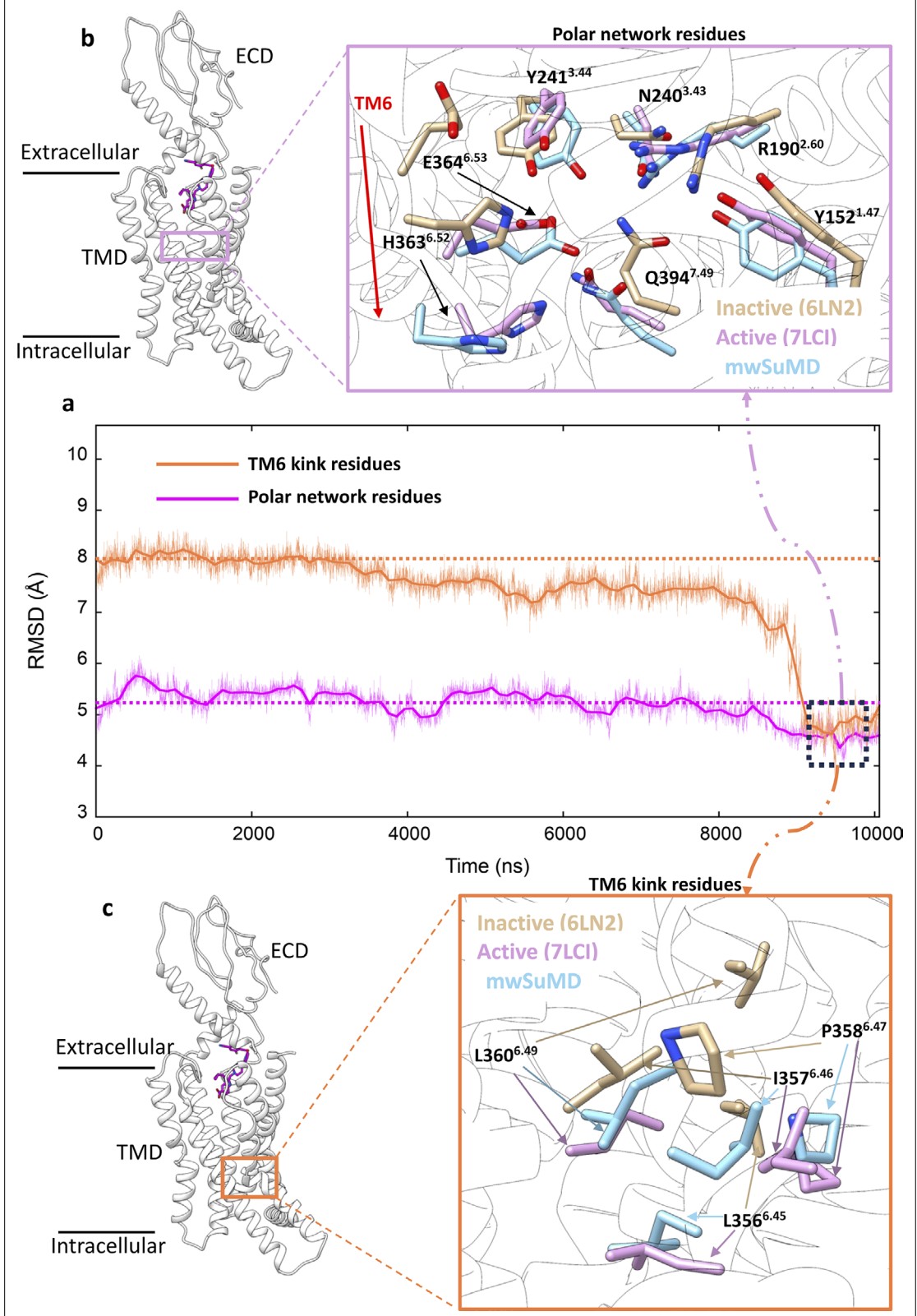

**Figure 4.** Glucagon-like peptide-1 receptor (GLP-1R) key structural motifs during multiple walker supervised molecular dynamics (mwSuMD) GLP-1R activation. (**a**) Root mean square deviation (RMSD) to the active state GLP-1R (7LCI) of the residues forming the central polar network (magenta) and TM6 kink (orange) during mwSuMD of receptor activation; at the end of the simulations minimum values were reached (dashed square). (**b**) The position of the polar network within the core of transmembrane domain (TMD) (left-hand panel) and comparison between inactive, active, and mwSuMD final

*Figure 4 continued on next page*

*Figure 4 continued*

states for the side chains of the residues forming of the polar network. (**c**) The position of the TM6 kink (right-hand panel) and comparison between inactive, active, and mwSuMD final states for the side chains of the residues forming the TM6 kink.

## Discussion

Classic MD simulations sample phase space with an efficiency that depends on the energy barrier between neighboring minima. Processes like (un)binding and protein activation require the system to overcome numerous energy barriers, some creating bottlenecks that slow the transition down to the millisecond, or second, time scale. To reduce some of these limits, we have developed and tested on complex structural events characterizing GPCRs, an energetically unbiased adaptive sampling algorithm, namely mwSuMD, which is based on traditional SuMD, while drawing on parallel multiple replica methods (*Dickson and Brooks, 2014*; *Sugita and Okamoto, 1999*).

Our simulations propose that remarkable predictivity can be obtained with distance-driven mwSuMD, as demonstrated by the lowest deviation from the experimental AVP:V$_2$R complex. The dissociation of AVP from V$_2$R was simulated much more rapidly by mwSuMD than by SuMD, suggesting it is an efficient tool for studying the dissociation of ligands from GPCRs. This is due to the more extensive sampling obtainable by seeding multiple parallel short simulations instead of a single simulation for batch.

mwSuMD performed similarly to SuMD for the dynamic docking of AVP to V2R when time windows of 600 ps were employed. Time windows of 100 ps remarkably improved mwSuMD. Usually, dynamic docking is performed to either predict the geometry of complexes or sample the binding path of an already-known intermolecular complex, or both. The RMSD of AVP to the experimental coordinates as the supervised metric produced the best results. Consequently, the RMSD should be the metric of choice to study the binding path of well-known intermolecular complexes. The distance, on the other hand, is necessary when limited structural information about the binding mode is available. In the absence of structural information regarding the final bound state, it is possible to sample numerous binding events employing mwSuMD and evaluate the final bound states rank by applying end-point free energy binding methods like the molecular mechanics energies combined with the Poisson-Boltzmann or generalized Born and surface area continuum solvation (MM/PBSA and MM/GBSA [*Wang et al., 2019*]) models.

We increased the complexity of binding simulations by considering GLP-1R and the non-peptide agonist PF06882961. Using mwSuMD, we obtained a binding of PF06882961 in good agreement with the cryo-EM structure, followed by an active-like conformational transition of GLP-1R. The choice of the metrics supervised was driven by the structural data available (*Zhang et al., 2020*) and extensive preparatory MD simulations. However, binding routes are possible from either the bulk solvent or the membrane (*Deganutti et al., 2021b*; *Stanley et al., 2016*; *Bokoch et al., 2018*). These results show the power of the mwSuMD method, indicating that future applications could include ligand binding

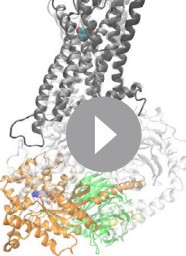

**Video 5.** G$_s$ binding to β$_2$ adrenoreceptor (β$_2$ AR). The inactive G$_s$ (G$_α$ subunit in orange, G$_β$ subunit in green, and G$_γ$ subunit in yellow) recognizes β$_2$AR (black ribbon) bound to epinephrine (van der Waals spheres). The experimental β$_2$AR:G$_s$ cryo-EM complex is reported in white ribbon for reference.

https://elifesciences.org/articles/96513/figures#video5

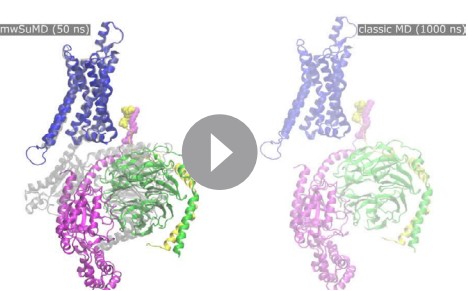

**Video 6.** G$_i$ binding to A1 receptor (A$_1$R). The G$_i$ (G$_α$ subunit in magenta, G$_β$ subunit in green, and G$_γ$ subunit in yellow) recognizes A$_1$R (blue ribbon) bound to adenosine (not shown). The experimental A$_1$R:G$_i$ cryo-EM complex is reported in transparent ribbon for reference.

https://elifesciences.org/articles/96513/figures#video6

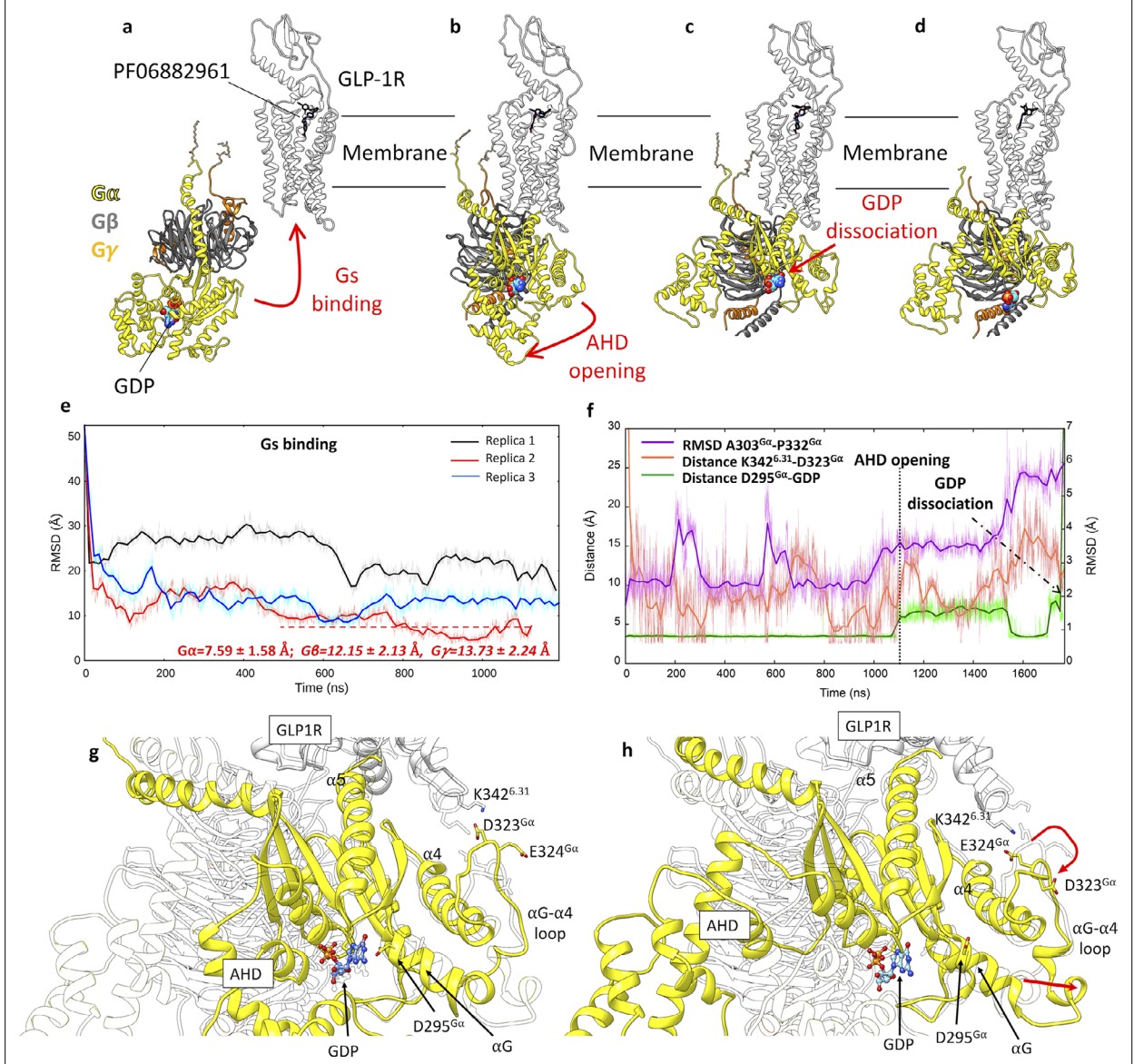

**Figure 5.** Glucagon-like peptide-1 receptor (GLP-1R) activation and Gs binding. (**a–d**) Sequence of simulated events during the multiple walker supervised molecular dynamics (mwSuMD) Gs:GLP-1R simulations. (**e**) Root mean square deviation (RMSD) of Gsα to the experimental GLP-1R:Gs complex (PDB 7LCJ) during three mwSuMD replicas; the RMSD to the experiential bound conformation (7LCJ) during the second part of Replica 2 (red dashed line) is reported for each Gs subunit. RMSDs were computed on Gα residues 11–43 and 205–394 to the experimental structure 7LCI after superimposition on GLP-1R residues 140–240 Cα atoms. (**f**) RMSD of the αG-α4 loop (purple), the distance between K342$^{6.31}$ and D323$^{Gα}$ (salmon), and the distance between guanosine diphosphate (GDP) and D295$^{Gα}$ (green) during Gs binding, alpha-helical domain (AHD) opening and GDP dissociation; (**g**) and (**h**) comparison between states extracted from before and after AHD opening. Before AHD opening (**a**), GLP-1R ICL3 interacted with D323$^{Gα}$ and D295$^{Gα}$ interacted with GDP; after AHD opening (**b**) and αG-α4 loop reorganization (curved red arrow), αG and D295$^{Gα}$ moved away from GDP (straight red arrow), destabilizing its binding to Gs.

The online version of this article includes the following figure supplement(s) for figure 5:

**Figure supplement 1.** G protein binding simulations to β2 adrenoreceptor (β2 AR) and A1 receptor (A1R).

**Figure supplement 2.** Comparison of the inactive (6EG8, white), nucleotide-free (7LCI, black), and guanosine diphosphate (GDP)-dissociated (multiple walker supervised molecular dynamics [mwSuMD] simulation, green) β6-α5 loop.

from the membrane, or alternative apo receptor conformations to improve the sampling for more difficult receptors.

mwSuMD enabled us to simulate the Gs binding to the active GLP-1R and the subsequent GDP release. Our results suggest a concerted effect on GDP binding produced by AHD opening, αG-α4

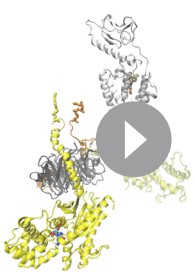

**Video 7.** G$_s$ binding to glucagon-like peptide-1 receptor (GLP-1R) and guanosine diphosphate (GDP) release. The Gs protein binds GLP-1R (light grey) in complex with PF06882961 (stick representation) before the alpha-helical domain opens and allows GDP release from G$_{s\alpha}$. The G$_{s\alpha}$ subunit is yellow, G$_\beta$ is dark gray, and G$_\gamma$ is orange.

https://elifesciences.org/articles/96513/figures#video7

loop rearrangement, and αG shift away from the GDP site. The full rotation and elongation of α5 as in cryo-EM structures would occur after the GDP release, supporting the role of hidden, metastable interactions as the driving force of G protein coupling and selectivity, as per recent work on GLP-1R (*Yuan et al., 2023*).

We stress that a complete understanding of a complex molecular event like G protein coupling requires the collection of numerous G protein binding paths and GDP dissociation events. mwSuMD is designed to yield a mechanistic description of structural events. For this reason, it integrates well with mutagenesis and kinetics experiments. We note that mwSuMD trajectories, since they describe the sequential states along a transition pathway, can represent a precious backbone for further MD sampling aimed at quantifying or predicting the kinetics of the transition. Approaches such as path collective variable metadynamics (*Alba Ortíz et al., 2019*), Markov state models (*Husic and Pande, 2018*), or machine learning models (*Kokh et al., 2019*; *Pérez et al., 2018*) can be informed by mwSuMD. We will address these novel opportunities created by mwSuMD in future work.

In summary, we showcased the applicability domain of mwSuMD to key, but scarcely understood, aspects of GPCR structural biology, pharmacology, and drug design hitherto unaddressed by unbiased simulations. Given the generality and simplicity of its implementation, we anticipate that mwSuMD can be employed to study a wide range of structural phenomena characterizing both membrane and cytosolic proteins. mwSuMD is an ongoing project updated on https://github.com/pipitoludovico/mwSuMD_multiEngine (copy archived at *Pipitò, 2025*) and, in the recent implementation, can exploit ACEMD (*Harvey et al., 2009*), NAMD (*Phillips et al., 2005*), GROMACS (*Berendsen et al., 1995*), and OPENMM (*Eastman et al., 2017*) as graphic processing units (GPU)-based MD engines while offering the option to run also on CPUs.

## Methods
### mwSuMD protocol

The SuMD is an adaptive sampling method (*Deganutti and Moro, 2017*) based on a tabu-like algorithm for speeding up the simulation of binding events between small molecules (or peptides [*Salmaso et al., 2017*; *Bower et al., 2018*]) and proteins (*Cuzzolin et al., 2016*; *Sabbadin and Moro, 2014*) without the introduction of any energetic bias. Briefly, during SuMD, a series of short unbiased MD simulations are performed, and after each simulation, the distances between the centers of mass (or the geometrical centers) of the ligand and the predicted binding site (collected at regular time intervals) are fitted to a linear function. If the resulting slope is negative (showing progress toward the target), the next simulation step starts from the last set of coordinates and velocities, otherwise, the simulation is restarted by randomly assigning the atomic velocities.

mwSuMD is designed to increase the sampling from a specific configuration by seeding user-decided parallel replicas (walkers) rather than one short simulation as per SuMD. Since one replica for each batch of walkers is always considered productive, mwSuMD gives more control than SuMD on the total wall-clock time used for a simulation. On the flip side, to maximize mwSuMD, it is optimal to assign one walker per GPU, requiring multiple GPUs to be effective. However, modern multi-threaded GPUs can still employ mwSuMD with a smaller cost in GPU performance. In the implementation for ACEMD used in this work, mwSuMD needs as input the initial coordinates of the system as a pdb file, the coordinates, and the atomic velocities of the system from the equilibration stage, the topology file of the system, and all the necessary force field parameters. The user can decide to supervise one (X) or two metrics (X′, X″) of the simulated system over short simulations seeded in batches, called

walkers. In the former case, either the slope of the linear function interpolating the metric values or a score can be adopted to decide whether to continue the mwSuMD simulation. When the user decides to supervise two metrics, a specific score is used. In the present work, distances between centroids, RMSDs, or the number of atomic contacts between two selections were supervised (*Table 1*). The choice of the metrics is system and problem-dependent, as the RMSD is most useful when the final state is known, while the distance is required when the target state is unknown; details on the scores are given below. The decision to restart or continue mwSuMD after any short simulation is postponed until all the walkers of a batch are collected. The best short simulation is selected and extended by seeding the same number of walkers, with the same duration as the step before.

For each walker, the score for the supervision of a single metric (SMscore) is computed as the square root of the product between the metric value in the last frame ($X_{last frame}$) and the average metric value over the short simulation ($\bar{X}$):

$$SMscore = \sqrt{X_{lastframe} * \bar{X}} \tag{1}$$

If the metric is set to decrease (e.g. binding or dimerization) the walker with the lowest SMscore is continued, otherwise (e.g. unbinding or outward opening of domains), the walker with the highest score is continued. Using the SMscore rather than the slope should give more weight to the final state of each short simulation, as it is the starting point for the successive batch of simulations. Considering the average of the metric should favor short simulations consistently evolving in the desired direction along the metric.

If both X' and X'' are set to increase during the mwSuMD simulations, the score for the supervision of two metrics (DMscore) on each walker is computed as follows:

$$DMscore = \left( \left( \frac{X'_{lastframe}}{\bar{X}'_{batchwalkers}} - 1 \right) + \left( \frac{X''_{lastframe}}{\bar{X}''_{batchwalkers}} - 1 \right) \right) * 100 \tag{2}$$

where $X'_{last frame}$ and $X''_{last frame}$ are the metrics values in the last frame, while $\bar{X}'_{batch walkers}$ and $\bar{X}''_{batch walkers}$ represent the average value of the two metrics over all the walkers in the batch. Subtracting the value 1 to the metric ratio ensures that if one of the two metrics from the last frame ($X'_{last frame}$ or $X''_{last frame}$) is equal to the average ($\bar{X}'_{batch walkers}$ or $\bar{X}''_{batch walkers}$), then that metric addend is null, and DMscore depends only on the remaining metric. If any of the two metrics is set to decrease, then the corresponding component in *Equation 2* is multiplied by –1 to maintain a positive score. Considering the average value of the two metrics over all the walkers rather than only over the considered walker should be more representative of the system evolution along the defined metric. In other words, the information about the metric is taken from all the walkers to better describe the evolution of the system.

The DMScore is designed to preserve some degree of independence between the two metrics supervised. Indeed, if the variation of one of them slows down and gets close to zero, the other metric is still able to drive the system's evolution. It should be noted that DMScore works at its best if the two metrics have similar variations over time, as in the case of distance and RMSD (both of which are distance-based). Notably, differently from SuMD, when a walker is extended by seeding a new batch of short simulations and the remaining walkers are stopped, the atomic velocities are not reassigned. This allows the simulations to be as short as a few picoseconds if desired, without introducing artifacts due to the thermostat latency to reach the target temperature (usually up to 10–20 ps when a simulation is restarted, reassigning the velocities of the atoms).

The current implementation of mwSuMD is for Python3 and exploits MDAnalysis (*Michaud-Agrawal et al., 2011*) and MDTRaj (*McGibbon et al., 2015*) modules.

## Force field, ligands parameters, and general systems preparation

The CHARMM36 (*Huang and MacKerell, 2013*; *Huang et al., 2017*) /CGenFF 3.0.1 (*Vanommeslaeghe and MacKerell, 2012*; *Vanommeslaeghe et al., 2012*; *Yu et al., 2012*) force field combination was employed in this work. Initial ligand force field, topology, and parameter files were obtained from the ParamChem webserver (*Vanommeslaeghe and MacKerell, 2012*). Restrained electrostatic potential (*Woods and Chappelle, 2000*) partial charges were assigned to all the non-peptidic small molecules but adrenaline and GDP using Gaussian09 (HF/6-31G* level of theory) and AmberTools20.

Six systems were prepared for MD (*Table 1*). Hydrogen atoms were added using the pdb2pqr (*Dolinsky et al., 2004*) and propka (*Olsson et al., 2011*) software (considering a simulated pH of 7.0); the protonation of titratable side chains was checked by visual inspection. The resulting receptors were separately inserted in a 1-palmitoyl-2-oleyl-*sn*-glycerol-3-phosphocholine (POPC) bilayer (previously built by using the VMD Membrane Builder plugin 1.1, Membrane Plugin, Version 1.1, at http://www.ks.uiuc.edu/Research/vmd/plugins/membrane/), through an insertion method (*Sommer, 2013*). Receptor orientation was obtained by superposing the coordinates on the corresponding structure retrieved from the OPM database (*Lomize et al., 2006*). Lipids overlapping the receptor TM helical bundle were removed and TIP3P water molecules (*Jorgensen et al., 1983*) were added to the simulation box employing the VMD Solvate plugin 1.5 (Solvate Plugin, Version 1.5. at https://www.ks.uiuc.edu/Research/vmd/plugins/solvate/). Finally, overall charge neutrality was reached by adding $Na^+/Cl^-$ counterions up to the final concentration of 0.150 M, using the VMD Autoionize plugin 1.3 (Autoionize Plugin, Version 1.3. at https://www.ks.uiuc.edu/Research/vmd/plugins/autoionize/).

## System equilibration and general MD settings

The MD engine ACEMD3 (*Harvey et al., 2009*) was employed for both the equilibration and productive simulations. The equilibration was achieved in isothermal-isobaric conditions (NPT) using the Berendsen barostat (*Berendsen et al., 1984*) (target pressure 1 atm) and the Langevin thermostat (*Loncharich et al., 1992*) (target temperature 300 K) with low damping of 1 $ps^{-1}$. For the equilibration (integration time step of 2 fs): first, clashes between protein and lipid atoms were reduced through 1500 conjugate-gradient minimization steps, then a positional constraint of 1 kcal $mol^{-1}$ $Å^{-2}$ on all heavy atoms was gradually released over different time windows: 2 ns for lipid phosphorus atoms, 60 ns for protein atoms other than alpha carbon atoms, 80 ns for alpha carbon atoms; a further 20 ns of equilibration was performed without any positional constraints.

Productive trajectories (*Table 1*) were computed with an integration time step of 4 fs in the canonical ensemble (NVT). The target temperature was set at 300 K, using a thermostat damping of 0.1 $ps^{-1}$; the M-SHAKE algorithm (*Forester and Smith, 1998*; *Kräutler et al., 2001*) was employed to constrain the bond lengths involving hydrogen atoms. The cutoff distance for electrostatic interactions was set at 9 Å, with a switching function applied beyond 7.5 Å. Long-range Coulomb interactions were handled using the particle mesh Ewald summation method (*Essmann et al., 1995*) by setting the mesh spacing to 1.0 Å.

## Vasopressin binding simulations

The $V_2R$ in complex with vasopressin (AVP) and the $G_s$ protein (*Zhou et al., 2021*) was retrieved from the Protein Data Bank (*Berman et al., 2000*) (PDB 7DW9). The $G_s$ was removed from the system and the missing residues on ECL2 (G185-G189) were modeled from scratch using Modeller 9.19 (*Fiser and Sali, 2003*), considering the solution with the lowest DOPE score out of 10 conformations produced. AVP was placed away from $V_2R$ in the extracellular bulk and the resulting system was prepared for MD simulations and equilibrated as reported above.

During SuMD simulations, the distance between the centroids of AVP residues C1-Q4 (backbone and side chains), anticipated to bind deep into $V_2R$, and the $V_2R$ residues lining the peptide binding site Q96, Q174, Q291, and L312 (Cα atoms only) was supervised over time windows of 600 ps or 100 ps (*Table 1*). mwSuMD simulations considered the same distance, the RMSD of AVP residues C1-Q4 to the experimental bound complex, or the combination of the two during time windows of 600 ps (3 walkers) or 100 ps (10 walkers) (*Table 1*). Slope, SMscore, or DMscore (see Methods section mwSuMD protocol) was used in the different mwSuMD replicas performed (*Table 1*). Simulations were stopped after 300 ns (time window duration = 600 ps) or 50 ns (time window duration = 100 ps) of total SuMD or mwSuMD simulation time.

## Vasopressin unbinding simulations

The $V_2R$:AVP complex was prepared for MD simulations and equilibrated as reported above. During both SuMD and mwSuMD simulations (*Table 1*), the distance between the centroids of AVP residues C1-Q4 (backbone and side chains) and $V_2R$ residues Q96, Q174, Q291, and L312 (Cα atoms only) was supervised over time windows of 100 ps (10 walkers seeded for mwSuMD simulations). Replicas were stopped when the AVP-$V_2R$ distance reached 40 Å.

## GLP-1R:PF06882961 binding simulations

The inactive GLP-1R was retrieved from the Protein Data Bank (*Berman et al., 2000*) (PDB 6LN2) (*Wu et al., 2020*). Fab and the intracellular negative allosteric modulator were removed, and missing residues in the stalk (129–134) and ICL2 (256–263) were modeled with Modeller 9.19, considering the solutions with the lowest DOPE score out of 10 conformations produced. The PF06882961 initial conformation was extracted from the complex with the fully active GLP-1R (*Zhang et al., 2021*) (PDB 7LCJ) and placed away from GLP-1R in the extracellular bulk. The resulting system was prepared for MD simulations and equilibrated as reported above. CGenFF dihedral force field parameters of PF06882961 with the highest penalties (dihedrals NG2R51-CG321-CG3C41-CG3C41 [penalty = 143.5] and NG2R51-CG321-CG3C41-OG3C51 [penalty = 152.4]) were optimized (*Figure 6a*) employing Gaussian09 (geometric optimization and dihedral scan at HF/6-31g(d) level of theory) and the VMD force field toolkit plugin (*Mayne et al., 2013*).

Four classic MD replicas, for a total of 8 μs, were performed on the inactive receptor (prepared for MD simulations and equilibrated as reported above) to assess the possible binding path to the receptor TMD and therefore decide the initial position of PF06882961 in the extracellular bulk of the simulation box. A visual inspection of the trajectories suggested three major conformational changes that could allow ligand access to the TMD (*Figure 5b and c*). Transitory openings of the ECD (distance Q47$^{ECD}$-S310$^{ECL2}$), TM6-TM7 (distance H363$^{6.52}$-F390$^{7.45}$), and TM1-ECL1 (distance E138$^{1.33}$ and W214$^{ECL1}$) were observed. Since the opening of TM1-ECL1 was observed in two replicas out of four, we placed the ligand in a favorable position for crossing that region of GLP-1R.

mwSuMD simulations (*Table 1*) were performed stepwise to dock the ligand within GLP-1R first and then relax the receptor toward the active state. The PF06882961 binding was obtained by super-vising at the same time the distance between the ligand's heavy atoms centroid and the centroid of GLP-1R TM7 residues L379$^{7.34}$-F381$^{7.36}$ (Cα atoms only), which are part of the orthosteric site, and the RMSD of the ECD (residues W33$^{ECD}$-W120$^{ECD}$, Cα atoms only) to the active state (PDB 7LCJ) until the former distance reached 4 Å. In the second phase of mwSuMD, the RMSD of the ECD (residues W33$^{ECD}$-W120$^{ECD}$, Cα atoms only) and the ECL1 to the active state (PDB 7LCJ, Cα atoms of residues M204$^{2.74}$-L224$^{3.27}$) were supervised until the latter reached less than 4 Å. During the third phase, the RMSDs of PF06882961, as well as the RMSD of ECL3 (residues A368$^{6.57}$-T378$^{7.33}$, Cα atoms), were supervised until the former reached values lower than 3 Å. In the last mwSuMD step, only the RMSD of TM6 (residues I345$^{6.34}$-F367$^{6.56}$, Cα atoms) to the active state (PDB 7LCJ) was supervised until less than 5 Å. RMSDs were computed after superimposition on TM2, ECL1, and TM3 residues 170–240 (Cα atoms), which is the GLP-1R less flexible part (*Deganutti et al., 2022*).

## Membrane-anchored G$_s$ protein:GLP-1R simulations and GDP dissociation

The PDB 6EG8 was processed through Charmm-GUI (*Jo et al., 2008*) to palmitoylate residue C3$^{Gαi}$ and geranylgeranylate residue C65$^{Gγ}$. The resulting system was inserted into a 120×120 Å POPC membrane and previously built by using the VMD Membrane Builder plugin 1.1, Membrane Plugin, Version 1.1, at http://www.ks.uiuc.edu/Research/vmd/plugins/membrane. Lipids overlapping the palmitoyl and geranylgeranyl groups were removed and TIP3P water molecules (*Jorgensen et al., 1983*) were added to the simulation box by means of the VMD Solvate plugin 1.5 (Solvate Plugin, Version 1.5. at https://www.ks.uiuc.edu/Research/vmd/plugins/solvate/). Finally, overall charge neutrality was reached by adding Na$^+$/Cl$^-$ counterions up to the final concentration of 0.150 M, using the VMD Autoionize plugin 1.3 (Autoionize Plugin, Version 1.3. at https://www.ks.uiuc.edu/Research/vmd/plugins/autoionize/). The first stage of equilibration was performed as reported above (Methods section System equilibration and general MD settings) for 120 ns, followed by a second stage in the NVT ensemble for a further 1 μs without any restraints to allow the membrane-anchored heterotrimeric G$_s$ protein to stabilize within the intracellular side of the simulation box. After this two-stage long equilibration, GLP-1R from the final frame of the activation simulation (in complex with PF06882961) was manually inserted into the equilibrated membrane above the G$_s$ protein using the corresponding structure retrieved from the OPM database as a reference, and the system further equilibrated for 120 ns as reported above (Methods section System equilibration and general MD settings). The GLP-1R-G$_s$ system was then subjected to three simulations (*Table 1*). Each mwSuMD replica was interrupted by 500 ns of classic MD twice, to relax the system during the transition. In the supervised stages, the

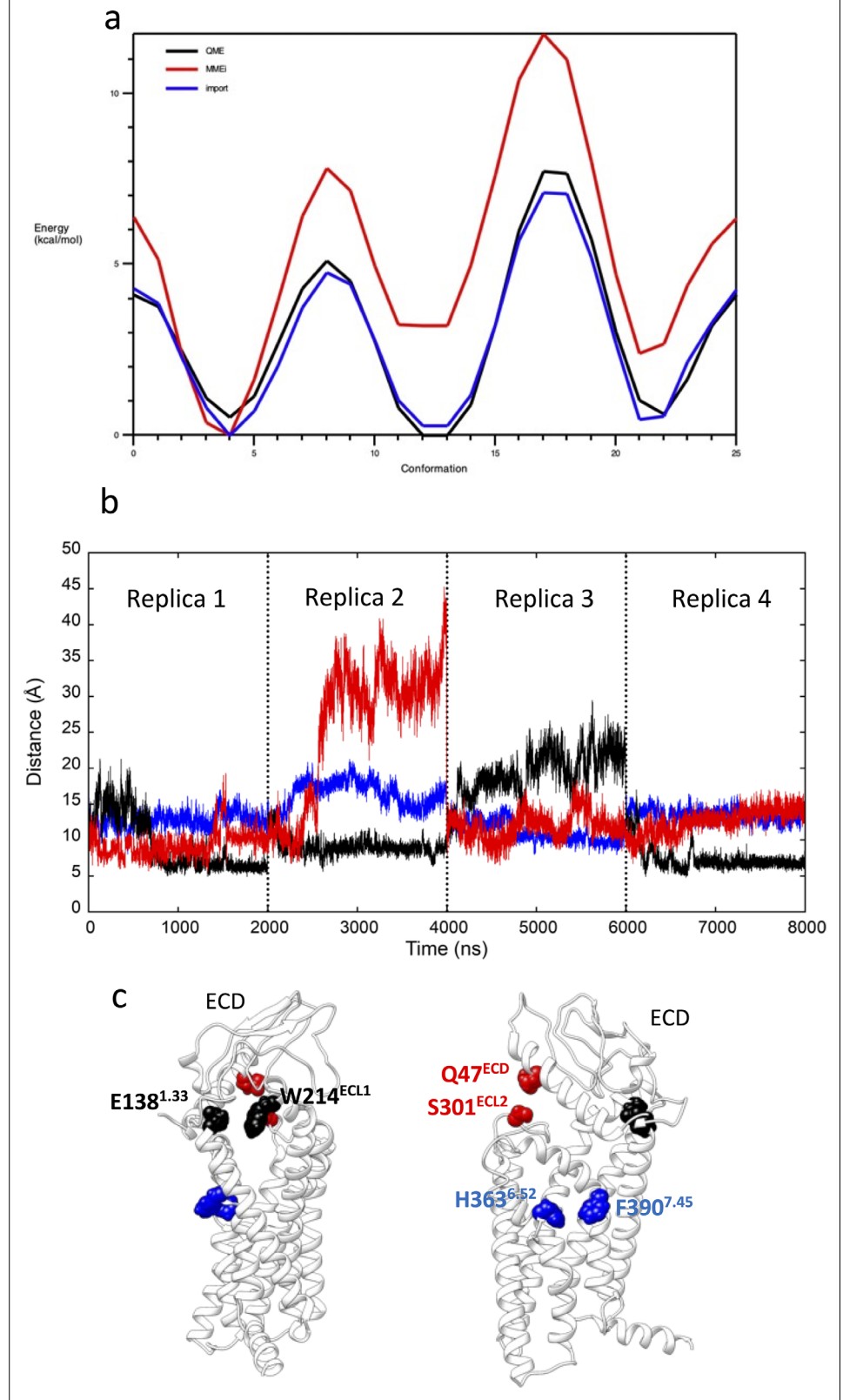

**Figure 6.** Preparation to GLP-1R mwSuMD simulations. (**a**) Potential energy surface derived from the scan of the PF06882961 rotatable bond through which the dihedrals NG2R51-CG321-CG3C41-CG3C41 (penalty = 143.5) and NG2R51-CG321-CG3C41-OG3C51 (penalty = 152.4) pass. The curve obtained from the original CGenFF parameters (red) was optimized (blue) to resemble the energy profile from quantum-mechanics computations

*Figure 6 continued on next page*

*Figure 6 continued*

at the HF/6-31g(d) level of theory (black). (**b**) Distances between glucagon-like peptide-1 receptor (GLP-1R) key residues E138$^{1.33}$ and W214$^{ECL1}$ (black), H363$^{6.52}$ and F390$^{7.45}$ (blue) and Q47$^{ECD}$ and S310$^{ECL2}$ (red) during 8 µs of classic molecular dynamics (MD); (**c**) positions of the three couples of residues reported as two-side views.

distance between residues M386-L394 $^{G\alpha s}$ (all-atoms centroid) of helix 5 (α5) and the GLP-1R intracellular residues R176$^{2.46}$, R348$^{6.37}$, S352$^{6.41}$, and N405$^{7.60}$ (Cα atoms only) was monitored, seeding 3 walkers of 200 ps each.

The AHD opening was simulated starting from the GLP-1R:G$_s$ binding mwSuMD replica with the final lowest G$_s$ RMSD, the lowest PF06882961 binding energy and the highest GDP binding energy (Replica 2 in *Figure 5e*) by supervising the distance between AHD residues G70-R199$^{G\alpha s}$ and K300-L394$^{G\alpha s}$ (all-atoms centroids) during 3 walkers of 100 ps each. 300 ns of classic MD was performed to relax the system. Finally, the GDP unbinding was supervised as the distance between GDP (all-atoms centroid) and residues E50$^{G\alpha s}$, K52$^{G\alpha s}$, T55$^{G\alpha s}$, K293$^{G\alpha s}$, and V367$^{G\alpha s}$ (Cα atoms only) of Gα$_s$; 5 walkers were used in a 50-ps-long mwSuMD simulations.

## MD analysis

Interatomic distances were computed through MDAnalysis *Michaud-Agrawal et al., 2011*; RMSD were computed using VMD (*Humphrey et al., 1996*) and MDAnalysis (*Michaud-Agrawal et al., 2011*). Interatomic contacts and ligand-protein hydrogen bonds were detected using the GetContacts scripts tool (https://getcontacts.github.io), setting a hydrogen bond donor-acceptor distance of 3.3 Å and an angle value of 120° as geometrical cutoffs. Contacts and hydrogen bond persistency are quantified as the percentage of frames (over all the frames obtained by merging the different replicas) in which protein residues formed contacts or hydrogen bonds with the ligand.

The MMPBSA.py (*Miller et al., 2012*) script, from the AmberTools20 suite (The Amber Molecular Dynamics Package, at http://ambermd.org/), was used to compute MM/GBSA method or the molecular mechanics MM/PBSA approach, after transforming the CHARMM psf topology files to an Amber prmtop format using ParmEd (documentation at https://parmed.github.io/ParmEd/html/index.html).

Supplementary videos were produced employing VMD and avconv (at https://libav.org/avconv.html). Molecular graphics images were produced using the UCSF Chimera (*Pettersen et al., 2004*) (Version 1.14).

## G$_s$ protein:β$_2$ AR binding simulations

mwSuMD simulations started from a not fully active, agonist-bound conformation of the β$_2$ AR and the inactive G$_s$ to resemble pre-coupling conditions. The full-length model of the adrenergic β$_2$ receptor (β$_2$ AR) in a not fully active state was downloaded from GPCRdb (https://gpcrdb.org/). The full agonist adrenaline (ALE) was inserted in the orthosteric site by superposition with the PDB ID 4LDO (fully active β$_2$ AR) (*Ring et al., 2013*). The structure of the inactive, GDP-bound G$_s$ protein (*Liu et al., 2019*) was retrieved from the Protein Data Bank (*Berman et al., 2000*) (PDB ID 6EG8) and placed in the intracellular bulk. The resulting system (G$_s$ >50 Å away from β$_2$ AR) was prepared for MD simulations and equilibrated as reported above. The PDB ID 3SN6 (fully active β$_2$ AR in complex with G$_s$ [*Rasmussen et al., 2011*]) was used as the reference for RMSD computations. Three mwSuMD replicas (*Table 1*) were performed supervising at the same time the distance between the helix 5 (α5) Gα$_s$ residues R385-L395 (Cα atoms centroid) and the β$_2$ AR (residues V31-P330 Cα atoms centroid) as well as the RMSD (superimposing on β$_2$ AR residues 70–170 Cα atoms) of β$_2$ AR TM6 residues C265-I278 (Cα atoms only) to the fully active state, during 100 ps time windows (5 walkers). To monitor the progression of the simulations, we computed the RMSD of the Cα atoms of the Gα (residues 11–43 and residues 205–394) and Gβ subunits (residues 3–340) to the experimental complex (*Rasmussen et al., 2011*; *Video 3*, *Figure 5—figure supplement 1*). The flexibility of G$_s$β is backed by both MD and cryo-EM data suggesting G protein rocking motions around G$_s$α:receptor interactions (*Dong et al., 2020*; *Liang et al., 2020*).

## Membrane-anchored G$_i$ protein:A$_1$R simulations

Since the full-length structure of the inactive human G$_i$ protein has not yet been resolved by X-ray or cryo-EM, it was modeled by superimposing the AlphaFold2 (*Jumper et al., 2021*) AI models of the

Gα$_i$ (P63096-F1), Gβ (Q9HAV0-F1), and Gγ (P50151-F1) subunits to the PDB file 6EG8 (a G$_s$ hetero-trimer). The resulting homotrimer (without GDP) was processed through Charmm-GUI (*Jo et al., 2008*) to palmitoylate residue C3$^{Gαi}$ and geranylgeranylate residue C65$^{Gγ}$ (*Linder et al., 1993*; *Mystek et al., 2019*). The side chains of these two lipidated residues were manually inserted into a 120×120 Å POPC membrane and previously built by using the VMD Membrane Builder plugin 1.1, Membrane Plugin, Version 1.1. at http://www.ks.uiuc.edu/Research/vmd/plugins/membrane/. Lipids overlapping the palmitoyl and geranylgeranyl groups were removed and TIP3P water molecules (*Jorgensen et al., 1983*) were added to the simulation box by means of the VMD Solvate plugin 1.5 (Solvate Plugin, Version 1.5. at https://www.ks.uiuc.edu/Research/vmd/plugins/solvate/). Finally, overall charge neutrality was reached by adding Na$^+$/Cl$^-$ counterions up to the final concentration of 0.150 M, using the VMD Autoionize plugin 1.3 (Autoionize Plugin, Version 1.3. at https://www.ks.uiuc.edu/Research/vmd/plugins/autoionize/). The first stage of equilibration was performed as reported above (Methods section System equilibration and general MD settings) for 120 ns, followed by a second stage in the NVT ensemble for a further 1 µs without any restraints to allow the membrane-anchored heterotrimeric G$_i$ protein to stabilize within the intracellular side of the simulation box. After this two-stage long equilibration, the active state A$_1$R in complex with adenosine (PDB 6D9H) was manually inserted into the equilibrated membrane above the G$_i$ protein using the corresponding structure retrieved from the OPM database as a reference, and the system further equilibrated for 120 ns as reported in the Methods section System equilibration and general MD settings. The A$_1$R-G$_i$ system was then subjected to a 1-µs-long classic MD simulation and an mwSuMD simulation (*Table 1*). During the mwSuMD simulation, the RMSD (superimposing on A$_1$R residues 40–140 Cα atoms) of helix 5 (α5) Gα$_i$ residues 329–354 to the PDB 6D9H was supervised, seeding 3 walkers of 100 ps each until the productive simulation time reached 50 ns (total simulation time 150 ns).

### Numbering system

Throughout the manuscript, the Ballesteros-Weinstein residues numbering system for class A (*Ballesteros and Weinstein, 1995*) and the Wootten residues numbering system for class B GPCRs (*Wootten et al., 2013*) are adopted.

## Acknowledgements

GD is a member of the GPCRs-focused European COST action ERNEST. CAR is grateful for a Royal Society Industry Fellowship. GD and CAR are grateful for support from the BBSRC (BB/W016974/1) and Diabetes UK (BDA 20/0006307).

## Additional information

### Funding

| Funder | Grant reference number | Author |
| --- | --- | --- |
| Biotechnology and Biological Sciences Research Council | BB/W016974/1 | Christopher Arthur Reynolds |
| Diabetes UK | BDA 20/0006307 | Giuseppe Deganutti |

The funders had no role in study design, data collection and interpretation, or the decision to submit the work for publication.

### Author contributions

Giuseppe Deganutti, Conceptualization, Data curation, Software, Formal analysis, Investigation, Visualization, Methodology, Writing – original draft, Writing – review and editing; Ludovico Pipito, Software, Investigation, Methodology, Writing – review and editing; Roxana Maria Rujan, Tal Weizmann, Peter Griffin, Investigation; Antonella Ciancetta, Stefano Moro, Methodology, Writing – review and editing; Christopher Arthur Reynolds, Conceptualization, Resources, Methodology, Writing – review and editing

## Author ORCIDs

Giuseppe Deganutti https://orcid.org/0000-0001-8780-2986
Ludovico Pipito https://orcid.org/0000-0002-1824-7541
Roxana Maria Rujan https://orcid.org/0000-0002-4335-3338
Peter Griffin https://orcid.org/0000-0003-0966-800X
Christopher Arthur Reynolds https://orcid.org/0000-0001-9267-5141

Reviewer #1 (Public review): https://doi.org/10.7554/eLife.96513.4.sa1
Reviewer #2 (Public review): https://doi.org/10.7554/eLife.96513.4.sa2
Reviewer #3 (Public review): https://doi.org/10.7554/eLife.96513.4.sa3
Author response https://doi.org/10.7554/eLife.96513.4.sa4

## Additional files

### Supplementary files
MDAR checklist

### Data availability
MD trajectories available at https://zenodo.org/records/7944479. mwSuMD software is available at https://github.com/pipitoludovico/mwSuMD_multiEngine (copy archived at *Pipitò, 2025*).

The following dataset was generated:

| Author(s) | Year | Dataset title | Dataset URL | Database and Identifier |
| --- | --- | --- | --- | --- |
| Deganutti G | 2023 | Updated MD trajectories for "Hidden GPCR structural transitions addressed by multiple walker supervised molecular dynamics (mwSuMD)" | https://doi.org/10.5281/zenodo.7944479 | Zenodo, 10.5281/zenodo.7944479 |

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

# Appendix

**Appendix 1—table 1.** GLP-1R:G$_s$ contacts during multiple walker supervised molecular dynamics (mwSuMD) G$_s$ binding simulations.

| GLP-1R residue | G$_s$ residue | Occupancy (% MD frames) |
| --- | --- | --- |
| ASP344 (6.33) | ARG385 | 87.7 |
| ARG348 (6.37) | LEU394 | 83 |
| GLU408 (7.63) | ARG356 | 64.4 |
| LEU254 (3.57) | TYR391 | 58.1 |
| ARG419 (7.83) | ASP312 | 55.3 |
| LYS415 (7.70) | ASP312 | 47.6 |
| TYR250 (3.53) | TYR391 | 46.3 |
| LEU254 (3.57) | LEU393 | 43.7 |
| ARG419 (7.83) | HIS311 | 43.2 |
| LEU254 (3.57) | LEU388 | 42.4 |
| VAL259 (ICL2) | ARG38 | 39.4 |
| SER352 (6.41) | LEU393 | 37.4 |
| LYS351 (6.40) | LEU394 | 36 |
| PHE257 (3.60) | GLN384 | 35.2 |
| GLU262 (4.38) | ARG38 | 35 |
| ARG348 (6.37) | ARG385 | 30.8 |
| ARG348 (6.37) | LEU393 | 30.6 |
| LEU251 (3.54) | LEU393 | 29.4 |
| ILE345 (6.34) | ARG385 | 23.8 |
| LEU349 (6.38) | LEU394 | 23.2 |
| HIS171 (ICL1) | ASP312 | 22.3 |
| LYS342 (6.31) | ASP323 | 21.6 |
| PHE260 (ICL2) | ARG38 | 21.5 |
| LEU254 (3.57) | GLN384 | 20.6 |
| ILE345 (6.34) | LEU394 | 19.9 |
| GLU262 (4.38) | TYR391 | 19,7 |
| SER261 (ICL2) | GLN35 | 19.7 |
| ARG348 (6.37) | ARG389 | 18.8 |
| LEU254 (3.57) | HIS387 | 18.6 |
| HIS171 (ICL1) | ASN313 | 17.6 |
| VAL259 (ICL2) | HIS387 | 16.8 |
| SER352 (6.41) | LEU394 | 16.7 |
| PHE257 (3.60) | HIS387 | 16.3 |
| LYS351 (6.40) | LEU393 | 16.3 |
| TYR250 (3.53) | LEU393 | 15.7 |
| LEU255 (3.58) | GLN384 | 15.6 |

*Appendix 1—table 1 Continued on next page*

*Appendix 1—table 1 Continued*

| GLP-1R residue | G$_s$ residue | Occupancy (% MD frames) |
|---|---|---|
| GLU412 (7.67) | ARG356 | 15 |
| LYS415 (7.70) | ASP354 | 14 |
| HIS173 (ICL1) | PHE335 | 13.9 |
| VAL259 (ICL2) | GLN35 | 13.8 |
| PHE257 (3.60) | LEU388 | 13.6 |
| HIS171 (ICL1) | ASP333 | 13.5 |
| SER258 (ICL2) | HIS387 | 13.1 |
| SER261 (ICL2) | ARG38 | 12.9 |
| ASN407 (7.62) | ARG356 | 12.5 |
| VAL259 (ICL2) | TYR391 | 12.4 |
| ARG170 (ICL1) | ASP312 | 12.3 |
| LEU255 (3.58) | LEU388 | 11.6 |
| LEU411 (7.66) | ARG356 | 11.5 |
| ARG176 (2.46) | GLU392 | 11.5 |
| GLN263 (4.39) | TRP332 | 11.5 |
| ILE345 (6.34) | LEU388 | 11.3 |
| HIS171 (ICL1) | ARG314 | 11.2 |
| VAL259 (ICL2) | ALA39 | 10.7 |
| TRP264 (4.40) | PHE335 | 10.5 |
| ARG267 (4.43) | ASP333 | 10.5 |
| HIS171 (ICL1) | TRP332 | 10.5 |
| PHE257 (3.60) | ARG38 | 10.2 |
| LEU349 (6.38) | LEU393 | 10 |
| GLN263 (4.39) | ASP333 | 10 |
| GLN263 (4.39) | ASN313 | 9.9 |
| ARG419 (7.83) | ASP354 | 9.7 |
| THR343 (6.32) | ARG385 | 9.1 |
| SER258 (ICL2) | ARG38 | 9 |
| LYS342 (6.31) | GLU322 | 8.7 |
| SER261 (ICL2) | ASP333 | 8.6 |
| LEU255 (3.58) | LEU394 | 8.6 |
| PHE260 (ICL2) | TYR391 | 8.5 |
| ASN406 (7.61) | GLU392 | 8.4 |
| SER258 (ICL2) | HIS41 | 8.4 |
| HIS171 (ICL1) | ARG337 | 8.4 |
| SER258 (ICL2) | GLN384 | 8.1 |
| ARG348 (6.37) | LEU388 | 7.9 |
| ARG267 (4.43) | ASN313 | 7.9 |

*Appendix 1—table 1 Continued on next page*

*Appendix 1—table 1 Continued*

| GLP-1R residue | G$_s$ residue | Occupancy (% MD frames) |
|---|---|---|
| HIS173 (ICL1) | ARG337 | 7.8 |
| PHE260 (ICL2) | GLN35 | 7.4 |
| ARG176 (2.46) | TYR391 | 7.3 |
| ARG267 (4.43) | PHE335 | 7.2 |
| ALA256 (3.59) | GLN384 | 7 |
| VAL259 (ICL2) | HIS41 | 7 |
| CYS174 (2.44) | ARG356 | 6.5 |
| GLU408 (7.63) | GLU392 | 6.1 |
| PHE257 (3.60) | TYR391 | 6.1 |
| SER261 (ICL2) | LYS34 | 6 |
| HIS173 (ICL1) | ARG314 | 6 |
| LEU251 (3.54) | LEU388 | 6 |
| HIS173 (ICL1) | ASP312 | 5.9 |
| HIS173 (ICL1) | TRP332 | 5.9 |
| VAL259 (ICL2) | LEU55 | 5.9 |
| ASN407 (7.62) | ARG389 | 5.8 |
| HIS171 (ICL1) | GLY310 | 5.8 |
| TYR178 (2.48) | ASP312 | 5.7 |
| SER258 (ICL2) | VAL217 | 5.7 |
| SER261 (ICL2) | GLN31 | 5.7 |
| LYS342 (6.31) | THR325 | 5.6 |
| LYS342 (6.31) | ALA324 | 5.5 |
| LEU254 (3.57) | LEU394 | 5.4 |
| CYS341 (6.30) | ASP323 | 5.4 |
| TRP264 (4.40) | ARG52 | 5.3 |
| GLU408 (7.63) | ARG389 | 5.2 |
| SER258 (ICL2) | TYR391 | 5.1 |

