## [Editor Report · eLife Assessment]

This study describes an improved adaptive sampling approach, multiple-walker Supervised Molecular Dynamics (mwSuMD), and its application to G protein-coupled receptors (GPCRs), which are the most abundant membrane proteins and key targets for drug discovery. The manuscript provides **solid** evidence that the mwSuMD approach can assist in the sampling of complex binding processes, leading to **useful** findings for GPCR activity, including resolution of interactions not seen experimentally. The method has the potential to have broad applicability in structural biology and pharmacology.

---

## [Referee Report · Reviewer #1 (Public review)]

Summary:

The authors investigate ligand and protein-binding processes in GPCRs (including dimerization) by the multiple walker supervised molecular dynamics method. The paper is interesting and it is very well written.

Strengths:

The authors' method is a powerful tool to gain insight on the structural basis for the pharmacology of G protein-coupled receptors.

---

## [Referee Report · Reviewer #2 (Public review)]

The study by Deganutti and co-workers is a methodological report on an adaptive sampling approach, multiple walker supervised molecular dynamics (mwSuMD), which represents an improved version of the previous SuMD.

Case-studies concern complex conformational transitions in a number of G protein Coupled Receptors (GPCRs) involving long time-scale motions such as binding-unbinding and collective motions of domains or portions. GPCRs are specialized GEFs (guanine nucleotide exchange factors) of heterotrimeric Gα proteins of the Ras GTPase superfamily. They constitute the largest superfamily of membrane proteins and are of central biomedical relevance as privileged targets of currently marketed drugs.

MwSuMD was exploited to address:

a) binding and unbinding of the arginine-vasopressin (AVP) cyclic peptide agonist to the V2 vasopressin receptor (V2R);

b) molecular recognition of the β2-adrenergic receptor (β2-AR) and heterotrimeric GDP-bound Gs protein;

c) molecular recognition of the A1-adenosine receptor (A1R) and palmotoylated and geranylgeranylated membrane-anchored heterotrimeric GDP-bound Gi protein;

d) the whole process of GDP release from membrane-anchored heterotrimeric Gs following interaction with the glucagon-like peptide 1 receptor (GLP1R), converted to the active state following interaction with the orthosteric non-peptide agonist danuglipron.

The revised version has improved clarity and rigor compared to the original also thanks to the reduction in the number of complex case studies treated superficially.

The mwSuMD method is solid and valuable, has wide applicability and is compatible with the most world-widely used MD engines. It may be of interest to the computational structural biology community.

The huge amount of high-resolution data on GPCRs makes those systems suitable, although challenging, for method validation and development.

While the approach is less energy-biased than other enhanced sampling methods, knowledge, at the atomic detail, of binding sites/interfaces and conformational states is needed to define the supervised metrics, the higher the resolution of such metrics is the more accurate the outcome is expected to be. Definition of the metrics is a user- and system-dependent process.

---

## [Referee Report · Reviewer #3 (Public review)]

Summary:

In the present work Deganutti et al. report a structural study on GPCR functional dynamics using a computational approach called supervised molecular dynamics.

Strengths:

The study has the potential to provide novel insight into GPCR functionality. An example is the interaction between D344 and R385 identified during the Gs coupling by GLP-1R. However, validation of the findings, even computationally through for instance in silico mutagenesis study, is advisable.

Weaknesses:

No significant advance of the existing structural data on GPCR and GPCR/G protein coupling is provided. Most of the results are reproductions of the previously reported structures.

---

## [Author Response]

The following is the authors’ response to the previous reviews

**Public Reviews**:
**Reviewer #1 (Public review):**
Summary:The authors investigate ligand and protein-binding processes in GPCRs (including dimerization) by the multiple walker supervised molecular dynamics method. The paper is interesting and it is very well written.Strengths:The authors' method is a powerful tool to gain insight on the structural basis for the pharmacology of G protein-coupled receptors.

We thank the Reviewer for the positive comment on the manuscript and the proposed methods.

**Reviewer #2 (Public review):**
The study by Deganutti and co-workers is a methodological report on an adaptive sampling approach, multiple walker supervised molecular dynamics (mwSuMD), which represents an improved version of the previous SuMD.Case-studies concern complex conformational transitions in a number of G protein Coupled Receptors (GPCRs) involving long time-scale motions such as binding-unbinding and collective motions of domains or portions. GPCRs are specialized GEFs (guanine nucleotide exchange factors) of heterotrimeric Gα proteins of the Ras GTPase superfamily. They constitute the largest superfamily of membrane proteins and are of central biomedical relevance as privileged targets of currently marketed drugs.MwSuMD was exploited to address:(a) binding and unbinding of the arginine-vasopressin (AVP) cyclic peptide agonist to the V2 vasopressin receptor (V2R);(b) molecular recognition of the β2-adrenergic receptor (β2-AR) and heterotrimeric GDPbound Gs protein;(c) molecular recognition of the A1-adenosine receptor (A1R) and palmotoylated and geranylgeranylated membrane-anchored heterotrimeric GDP-bound Gi protein;(d) the whole process of GDP release from membrane-anchored heterotrimeric Gs following interaction with the glucagon-like peptide 1 receptor (GLP1R), converted to the active state following interaction with the orthosteric non-peptide agonist danuglipron.The revised version has improved clarity and rigor compared to the original also thanks to the reduction in the number of complex case studies treated superficially.The mwSuMD method is solid and valuable, has wide applicability and is compatible with the most world-widely used MD engines. It may be of interest to the computational structural biology community.The huge amount of high-resolution data on GPCRs makes those systems suitable, although challenging, for method validation and development.While the approach is less energy-biased than other enhanced sampling methods, knowledge, at the atomic detail, of binding sites/interfaces and conformational states is needed to define the supervised metrics, the higher the resolution of such metrics is the more accurate the outcome is expected to be. Definition of the metrics is a user- and system-dependent process.

We thank the Reviewer for the positive comment on the revised manuscript and mwSuMD. We agree that the choice of supervised metrics is user- and systemdependent. We aim to improve this aspect in the future with the aid of interpretable machine learning.

**Reviewer #3 (Public review):**
Summary:In the present work Deganutti et al. report a structural study on GPCR functional dynamics using a computational approach called supervised molecular dynamics.Strengths:The study has potential to provide novel insight into GPCR functionality. Example is the interaction between D344 and R385 identified during the Gs coupling by GLP-1R. However, validation of the findings, even computationally through for instance in silico mutagenesis study, is advisable.Weaknesses:No significant advance of the existing structural data on GPCR and GPCR/G protein coupling is provided. Most of the results are reproductions of the previously reported structures.

The method focus of our study (mwSuMD) is an enhancement of the supervised molecular dynamics that allows supervising two metrics at the same time and uses a score, rather than a tabù-like algorithm, for handing the simulation. Further changes are the seeding of parallel short replicas (walkers) rather than a series of short simulations, and the software implementation on different MD engines (e.g. Acemd, OpenMM, NAMD, Gromacs).

We agree with the Reviewer that experimental validation of the findings would be advisable, in line with any computational prediction. We are positive that future studies from our group employing mwSuMD will inform mutagenesis and BRET-based experiments.

**Reviewer #2 (Recommendations for the authors):**
As for GLP1R, I remain convinced that the 7LCI would have been better as a reference for all simulations than 7LCJ, also because 7LCI holds a slightly more complete ECD.

We agree that 7LCJ would have been a better starting point than 7LCI for simulations because it presents the stalk region, contrary to 7LCJ. However, we do not think it might have influenced the output because the stalk is the most flexible segment of GLP1R, and any initial conformation is usually not retained during MD simulations.

Please, correct everywhere the definition of the 6LN2 structure of GPL1R as a ligand-free or apo, because that structure is indeed bound to a negative allosteric modulator docked on the cytosolic end of helix-6

We thank the reviewer for this precision. The text has been modified accordingly.

As for the beta2-AR, the "full-length" AlphaFold model downloaded from the GPCRdb is not an intermediate active state because it is very similar to the receptor in the 3SN6 complex with Gs. Please, eliminate the inappropriate and speculative adjective "intermediate".

We have changed “intermediate” to “not fully active”, which is less speculative since full activation can be achieved only in the presence of the G protein.

Incidentally, in that model, the C-tail, eliminated by the authors, is completely wrong and occupies the G protein binding site. It is not clear to me the reason why the authors preferred to used an AlphaFold model as an input of simulations rather than a high resolution structural model, e.g. 4LDO. Perhaps, the reason is that all ICL regions, including ICL3, were modeled by AlphaFold even if with low confidence. I disagree with that choice.

We understand the reviewer’s point of view. Should we have simulated an “equilibrium” receptor-ligand complex, we would have made the same choice. However, the conformational changes occurring during a G protein binding are so consistent that the starting conformation of the receptor becomes almost irrelevant as long as a sensate structure is used.

**Reviewer #3 (Recommendations for the authors):**
The revised version of the manuscript is more concise, focusing only on two systems. However, the authors have responded superficially to the reviewers' comments, merely deleting sections of text, making minor corrections, or adding small additions to the text. In particular, the authors have not addressed the main critical points raised by both Reviewer 2 and Reviewer 3.For example, the RMSD values for the binding of PF06882961 to GLP-1R remain high, raising doubts about the predictive capabilities of the method, at least for this type of system.What is the RMSD of the ligand relative to the experimental pose obtained in the simulations? This value must be included in the text.

We have added this piece of information about PF06882961 RMSD in the text, which on page 6 now reads “We simulated the binding of PF06882961, reaching an RMSD to its bound conformation in 7LCJ of 3.79 +- 0.83 Å (computed on the second half of the merged trajectory, superimposing on GLP-1R Ca atoms of TMD residues 150 to 390), using multistep supervision on different system metrics (Figure 2) to model the structural hallmark of GLP-1R activation (Video S5, Video S6).”

Similarly, the activation mechanism of GLP-1R is only partially simulated.Furthermore, it is not particularly meaningful to justify the high RMSD values of the SuMD simulations for the binding of Gs to GLP-1R by comparing them with those reported under unbiased MD conditions. "Replica 2, in particular, well reproduced the cryo-EM GLP-1R complex as suggested by RMSDs to 7LCI of 7.59{plus minus}1.58Å, 12.15{plus minus}2.13Å, and 13.73{plus minus}2.24Å for Gα, Gβ, and Gγ respectively. Such values are not far from the RMSDs measured in our previous simulations of GLP-1R in complex with Gs and GLP-149 (Gα = 6.18 {plus minus} 2.40 Å; Gβ = 7.22 {plus minus} 3.12 Å; Gγ = 9.30 {plus minus} 3.65 Å), which indicates overall higher flexibility of Gβ and Gγ compared to Gα, which acts as a sort of fulcrum bound to GLP-1R."Without delving into the accuracy of the various calculations, the authors should acknowledge that comparing protein structures with such high RMSD values has no meaningful significance in terms of convergence toward the same three-dimensional structure.

The text has been edited to accommodate the reviewer’s suggestion and still give the readers the measure of the high flexibility of Gs bound to GLP-1R. It now reads “Such values do not support convergence with the static experimental structure but are not far from the RMSDs measured in our previous simulations of GLP-1R in complex with G_s_ and GLP-1 (G_α_ = 6.18 ± 2.40 Å; G_b_ = 7.22 ± 3.12 Å; G_g_ = 9.30 ± 3.65 Å), which indicates overall higher flexibility of G_b_ and G_g_ compared to G_α_, which acts as a sort of fulcrum bound to GLP-1R.”

Have the authors simulated the binding of the Gs protein using the experimentally active structure of GLP-1R in complex with the ligand PF06882961 (PDB ID 7LCJ)? Such a simulation would be useful to assess the quality of the binding simulation of Gs to the GLP1R/PF06882961 complex obtained from the previous SuMD.

We considered performing the Gs binding simulation to the active structure of GLP-1R.

However, the GLP-1R (and other class B receptors) fully active state, as reported in 7LCJ, depends on the presence of the Gs and can be reached only upon effector coupling. Since it is unlikely that the unbound receptor is already in the fully active state, we reasoned that considering it as a starting point for Gs binding simulations would have been an artifact.

An example of the insufficient depth of the authors' replies can be seen in their response: "We note that among the suggested references, only Mafi et al report about a simulated G protein (in a pre-formed complex) and none of the work sampled TM6 rotation without input of energy."This statement is inaccurate. For instance, D'Amore et al. (Chem 2024, doi: 10.1016/j.chempr.2024.08.004) simulated Gs coupling to A2A as well as TM6 rotation, as did Maria-Solano and Choi (eLife 2023, doi: 10.7554/eLife.90773.1). The former employed path collective variables metadynamics, which is not cited in the introduction or the discussion, despite its relevance to the methodologies mentioned.

Respectfully, our previous reply is correct, as all of the mentioned articles used enhanced (energy-biased) approaches, so the claim “none of the work sampled TM6 rotation without input of energy” stands. The reference to D’Amore et al. (published after the previous round of reviews of this manuscript) has been added to the introduction; we thank the reviewer for pointing it out.

Additionally, SuMD employs a tabu algorithm that applies geometric supervision to the simulation, serving as an alternative approach to enhancing sampling compared to the "input of energy" techniques as called by the authors. A fair discussion should clearly acknowledge this aspect of the SuMD methodology.

We have now specified in the Methods that a tabù-like algorithm is part of SuMD, which, despite being the parent technique of mwSuMD, is not the focus of the present work. We provide extended references for readers interested in SuMD. mwSuMD, on the other hand, does not use a tabù-like algorithm but rather a continuative approach based on a score to select the best walker for each batch, as described in the Methods.